# Guarantees of a Preconditioned Subgradient Algorithm for Overparameterized Asymmetric Low-rank Matrix Recovery

Paris Giampouras [1]    HanQin Cai [2]    René Vidal [3]

## Abstract

In this paper, we focus on a matrix factorization-based approach to recover low-rank *asymmetric* matrices from corrupted measurements. We propose an *Overparameterized Preconditioned Subgradient Algorithm (OPSA)* and provide, for the first time in the literature, linear convergence rates independent of the rank of the sought asymmetric matrix in the presence of gross corruptions. Our work goes beyond existing results in preconditioned-type approaches addressing their current limitation, i.e., the lack of convergence guarantees in the case of *asymmetric matrices of unknown rank*. By applying our approach to (robust) matrix sensing, we highlight its merits when the measurement operator satisfies a mixed-norm restricted isometry property. Lastly, we present extensive numerical experiments that validate our theoretical results and demonstrate the effectiveness of our approach for different levels of overparameterization and outlier corruptions.

## 1. Introduction

Low-rank matrix recovery has been a ubiquitous problem showing up in numerous applications in the fields of signal/image processing, machine learning, and data science (Recht et al., 2010; Chen et al., 2013; Davenport & Romberg, 2016; Cai et al., 2021a; Smith et al., 2024; Wang et al., 2024). For instance, problems such as matrix sensing (Jain et al., 2013; Wei et al., 2016; Park et al., 2017; Li et al., 2020), matrix completion (Candes & Recht, 2008; Nie et al., 2012;

[1]Department of Computer Science, University of Warwick, Coventry, England, UK [2]Department of Statistics and Data Science and Department of Computer Science, University of Central Florida, Orlando, FL, USA [3]Center for Innovation in Data Engineering and Science (IDEAS), Departments of ESE and Radiology, University of Pennsylvania, Philadelphia, PA, USA. Correspondence to: René Vidal <vidalr@seas.upenn.edu>.

*Proceedings of the 42$^{nd}$ International Conference on Machine Learning*, Vancouver, Canada. PMLR 267, 2025. Copyright 2025 by the author(s).

Kümmerle & Verdun, 2021; Cai et al., 2023), and robust principal component analysis (Candès et al., 2011; Netrapalli et al., 2014; Giampouras et al., 2018; Cai et al., 2019; 2021b), can all be cast as low-rank matrix recovery problems and then solved using minimization algorithms that seek a matrix $X \in \mathbb{R}^{m \times n}$ that is as close as possible to the unknown low-rank matrix $X_\star \in \mathbb{R}^{m \times n}$.

A major challenge in low-rank matrix recovery concerns the computational complexity and memory requirements of the proposed algorithms when the size of the unknown matrix significantly increases. To address this, matrix factorization-based approaches have been proposed, (Chi et al., 2019; Park et al., 2016), which, given $r \geq \text{rank}(X_\star)$, use matrices $L \in \mathbb{R}^{m \times r}$ and $R \in \mathbb{R}^{n \times r}$ such that $X = LR^\top$. These approaches enable the development of iterative algorithms with significantly reduced computational complexity and memory storage requirements. However, they lead to non-convex formulations of the optimization problems, which pose their own challenges in the derivation of theoretical guarantees and fast rates of convergence.

Alternating gradient-based algorithms have been a standard approach for solving matrix factorization-based problems. A well-known issue with these approaches lies in the dependence of their rate of convergence on the condition number of the unknown matrix $X_\star$,(Tong et al., 2021a). This makes convergence too slow in problems where the sought matrix is ill-conditioned. Several works, (Park et al., 2016; Zhang et al., 2023), address this issue by moving beyond vanilla gradient updates and resorting to *preconditioned* approaches. In (Tong et al., 2021a), preconditioned gradient algorithms have been proposed that consist of updates in the following form:

$$L_{t+1} = L_t - \eta \nabla_L \mathcal{L}(L_t R_t^\top) \underbrace{(R_t^\top R_t)^{-1}}_{\text{preconditioner}},$$

$$R_{t+1} = R_t - \eta \nabla_R \mathcal{L}(L_t R_t^\top) \underbrace{(L_t^\top L_t)^{-1}}_{\text{preconditioner}}. \tag{1}$$

Preconditioned-based approaches, e.g. (Chi et al., 2019; Cai et al., 2021c; Zhang et al., 2023; 2024; Cai et al., 2024), lead to linear rates of convergence that do not depend on the condition number of $X_\star$. Recently, these approaches have been extended to the *robust low-rank matrix recovery*

framework, aiming to address more challenging scenarios that account for the presence of grossly corrupted data. In these settings, (Tong et al., 2021b) reported similar rates of convergence, which are however negatively influenced as the level of corruption increases.

A fundamental assumption made in the above-mentioned approaches is that the rank $r$ of the sought matrix $\boldsymbol{X}_\star$ is known. Clearly, in the case that $r$ is underestimated, the algorithm will only be able to find a low-rank approximation of the ground truth matrix. To address this shortcoming, in (Ma & Fattahi, 2021), the authors overparameterized the rank as $d \geq r = \text{rank}(\boldsymbol{X}_\star)$, and established a linear rate of convergence for robust low-rank matrix recovery for *symmetric* matrices using a vanilla subgradient algorithm. Further works, build on this observation aiming to establish convergence to the ground truth matrix $\boldsymbol{X}_\star$ under weaker conditions, (Ding et al., 2021; Xiong et al., 2023), or to explore the intriguing implicit regularization phenomena in this setting, (Soltanolkotabi et al., 2023).

Note that due to non-invertibility issues as the iterates $\boldsymbol{L}_t, \boldsymbol{R}_t$ converge to $\boldsymbol{L}_\star, \boldsymbol{R}_\star$, the overparameterized scenario cannot be directly adopted as such in the previously proposed preconditioned-based methods that rely on updates given in (1). To address this issue, recent works, (Zhang et al., 2023; Xu et al., 2023), proposed an overparameterized preconditioned algorithm focusing on symmetric matrices and assuming smooth loss functions, with updates in the form:

$$\boldsymbol{L}_{t+1} = \boldsymbol{L}_t - \eta\nabla_{\boldsymbol{L}}\mathcal{L}(\boldsymbol{L}_t\boldsymbol{L}_t^\top)(\boldsymbol{L}_t^\top\boldsymbol{L}_t + \lambda\boldsymbol{I})^{-1}, \quad (2)$$

and showed linear convergence at a rate independent of the condition number of $\boldsymbol{X}_\star$ and the overparameterization of the true rank. Focusing again on *smooth* problems, in (Cheng & Zhao, 2024), the authors proposed an extension of these works to the case of asymmetric matrices coming up with an alternating algorithm with regularized preconditioners leading to updates:

$$\begin{aligned} \boldsymbol{L}_{t+1} &= \boldsymbol{L}_t - \eta\nabla_{\boldsymbol{L}}\mathcal{L}(\boldsymbol{L}_t\boldsymbol{R}_t^\top)(\boldsymbol{R}_t^\top\boldsymbol{R}_t + \lambda_t\boldsymbol{I})^{-1}, \\ \boldsymbol{R}_{t+1} &= \boldsymbol{R}_t - \eta\nabla_{\boldsymbol{R}}\mathcal{L}(\boldsymbol{L}_t\boldsymbol{R}_t^\top)(\boldsymbol{L}_t^\top\boldsymbol{L}_t + \lambda_t\boldsymbol{I})^{-1}. \end{aligned} \quad (3)$$

In this work, we depart from previous works by focusing on *robust* low-rank matrix recovery in the presence of outliers and propose an overparameterized preconditioned-based algorithm in the unknown rank regime. Our work aims to address the following question (Q):

> *Q: Can we establish linear rates of convergence to the ground truth $\boldsymbol{X}_\star$, in the case of **non-smooth** minimization problems in the overparameterized regime with **unknown rank**, and for **asymmetric** matrices $\boldsymbol{X}_\star$?*

## 1.1. Main Contributions

In this work, we advance beyond previous work by focusing on robust low-rank matrix recovery with a non-smooth objective function, addressing the unique challenges of recovering *asymmetric matrices with unknown rank* (see Table 1 and comparison with SOTA). Our preconditioners naturally arise by adopting quasi-Newton-type updates in an implicitly regularized objective function. Our main contributions are summarized as follows:

- We propose a novel algorithm, coined *Overparameterized Preconditioned Subgradient Algorithm (OPSA)*, that minimizes a robust $\ell_1$ loss function. To account for overparameterization caused by rank overestimation, we propose a novel distance metric and assume that the matrix factors are initialized sufficiently close to the ground truth (which can be easily attained by spectral initialization). In Theorem 5.4, we show that *OPSA converges linearly* to the low-rank ground truth matrix $\boldsymbol{X}_\star$ using an adaptive Polyak's step size. Note that our main result holds for general non-smooth loss functions under certain conditions such as the restricted rank-$d$ sharpness condition and restricted Lipschitz continuity. Moreover, our results extend the preconditioned subgradient method (Tong et al., 2021b) from the exact known rank setting to the overparameterized regime.

- For theoretical results, we focus on robust matrix sensing and show that linear convergence holds for OPSA both in the noiseless case and in the presence of gross corruptions/outliers when the measurement matrices satisfy a mixed-norm restricted isometry property (RIP). In this setting, we unveil how the iteration complexity is affected by overparameterization. Moreover, our results showcase that the tolerance of measurement matrices in outliers is another important factor for sharpness around $\boldsymbol{X}_\star$, which is a necessary condition for exact convergence.

- In the experimental section, we empirically showcase the favorable performance of the proposed OPSA against the state-of-the-art under different levels of overparameterization $d$, for the problem of robust matrix sensing with Gaussian measurements. We also demonstrate that OPSA constantly enjoys linear convergence with varying condition numbers $\kappa$, parameters $\lambda$, and outlier densities even when the rank is heavily overestimated. The experiments provided further confirm our theoretical findings.

## 1.2. Notation

The transpose of a vector or matrix is denoted as $(\cdot)^\top$. The Euclidean vector norm is denoted as $\|\cdot\|_2$. The Frobenius and operator matrix norms are denoted as $\|\cdot\|_F$ and $\|\cdot\|_{op}$,

*Table 1.* Comparison of theoretical convergence properties of SOTA algorithms for low-rank matrix estimation. Columns indicate if algorithms handle asymmetric matrices, outliers, unknown rank, and if convergence is independent of $\kappa(\boldsymbol{X}_\star)$, the condition number of $\boldsymbol{X}_\star$. The proposed OPSA method addresses all these challenges.

| Algorithm | Asymmetric | Outliers | Unknown Rank | No dependency on $\kappa(\boldsymbol{X}_\star)$ |
|---|---|---|---|---|
| VANILLA GD (Stöger & Soltanolkotabi, 2021) | ✓ | ✓ | ✓ | ✗ |
| SCALEDGD (Tong et al., 2021a) | ✓ | ✗ | ✗ | ✓ |
| PRECGD (Zhang et al., 2023) | ✗ | ✗ | ✓ | ✓ |
| SCALEDGD($\lambda$) (Xiong et al., 2023) | ✗ | ✗ | ✓ | ✓ |
| SCALEDSM (Tong et al., 2021b) | ✓ | ✓ | ✗ | ✓ |
| OPSA (Proposed) | ✓ | ✓ | ✓ | ✓ |

respectively. We also denote as $\sigma_i(\boldsymbol{X})$ the singular values of matrix $\boldsymbol{X}$ -assuming a decreasing order as $i$ increases- and as $\kappa(\boldsymbol{X})$ its condition number. We denote the trace of a matrix as $\mathrm{tr}(\cdot)$, the trace of the inner product between two matrices $\boldsymbol{A}, \boldsymbol{B}$ as $\langle \boldsymbol{A}, \boldsymbol{B} \rangle = \mathrm{tr}(\boldsymbol{A}^\top \boldsymbol{B})$, and the $m \times n$-dimensional Euclidean space as $\mathbb{R}^{m \times n}$. $G(d)$ denotes the set of invertible matrices in $\mathbb{R}^{d \times d}$.

## 2. Related Work

In this section, we provide some interesting insights into the connection of our approach with prior research works.

**Preconditioned gradient and subgradient methods for low-rank matrix recovery.** Preconditioned-based methods have attracted significant interest over the last few years since they allow for establishing rates of convergence that do not depend on the condition number of $\boldsymbol{X}_\star$, (Tong et al., 2021a; Zhang, 2021; Zhang et al., 2023). For a thorough review of these methods, we refer the reader to (Chi et al., 2019). Preconditioned methods have been extended to non-smooth problems such as robust low-rank matrix recovery with $\ell_1$ loss. However, they focus on either the known-rank asymmetric regime or assume an unknown-rank with sought symmetric matrix $\boldsymbol{X}$. Relaxing the symmetric assumption on $\boldsymbol{X}$ to the more challenging asymmetric one in unknown-rank regimes is the main contribution of our work. Recently, these approaches have been shown to offer significant improvements in the low-rank adaptation (LoRA) for parameter-efficient fine-tuning foundation models, (Zhang & Pilanci, 2024). Even though this problem is out of the scope of the current paper, extending current approaches, which rely on a fixed rank, to the overparameterized preconditioning framework is a promising future research direction.

**Overparameterized (robust) low-rank matrix recovery.** Recently, several works have focused on robust low-rank matrix recovery in the unknown rank regime. In (Ma & Fattahi, 2021), the authors focus on robust matrix sensing and report the convergence of a vanilla subgradient algorithm in the overparameterized setting for symmetric matrices, which suggests an implicit regularization behavior. In (Ding et al., 2021), improved results are obtained, again for the symmetric case, by relaxing the conditions imposed on measurement matrices. In (Zhang et al., 2023), with the aim to reduce the negative effect of overparameterization and ill-conditioning, the authors focused on symmetric matrices and generalized the preconditioned-based approach in the overparameterized setting using updates in the form of (2). Similar to our work, the authors in (Cheng & Zhao, 2024) recently, proposed an overparameterized preconditioned approach for asymmetric matrix factorization establishing linear convergence with update in the form of (3). However, unlike our work, they focused on smooth losses, which pose less challenges, and enabled them to use a Polyak-Lojasiewicz (PL)-type condition for deriving the convergence rate.

## 3. Problem Formulation

We focus on the low-rank matrix estimation problem, assuming that the true rank $r$ is unknown. We denote the ground truth matrix as $\boldsymbol{X}_\star$, and assume a singular value decomposition

$$\boldsymbol{X}_\star = \boldsymbol{U}_\star \boldsymbol{\Sigma}_\star \boldsymbol{V}_\star^\top, \qquad (4)$$

where $\boldsymbol{U}_\star \in \mathbb{R}^{m \times d}$ contains $d \geq r$ left singular vectors, $\boldsymbol{\Sigma}_\star \in \mathbb{R}^{d \times d}$ is a diagonal matrix consisting of $d$ singular values of $\boldsymbol{X}_\star$ presented in an non-ascending order. Since $\mathrm{rank}(\boldsymbol{X}_\star) = r$ and $d \geq r$ we have $\sigma_i(\boldsymbol{X}_\star) = 0$ for $i = r+1, \ldots, d$.

The low-rank matrix estimation problem w.r.t. the space of $\boldsymbol{X} \in \mathbb{R}^{m \times n}$ is defined as

$$\min_{\substack{\boldsymbol{X} \in \mathbb{R}^{m \times n}, \\ \mathrm{rank}(\boldsymbol{X}) \leq d}} \mathcal{L}(\boldsymbol{X}), \qquad (5)$$

where $\mathcal{L}(\boldsymbol{X})$ is a general loss function that is convex w.r.t. $\boldsymbol{X}$ and possibly non-smooth in order to allow the use of robust loss functions such as the $\ell_1$ norm.

Here, we solve a problem equivalent to problem (5), defined

over matrix factors $L \in \mathbb{R}^{m \times d}$ and $R \in \mathbb{R}^{n \times d}$,

$$\min_{L \in \mathbb{R}^{m \times d}, R \in \mathbb{R}^{n \times d}} \mathcal{L}(LR^\top). \quad (6)$$

### 3.1. Matrix Sensing

Next, we focus on the *matrix sensing* problem i.e., we assume that we have access to observations $\mathbf{y} = \{y_i\}_{i=1}^p$ of a low-rank matrix $X_\star \in \mathbb{R}^{m \times n}$, given as

$$y_i = \mathcal{A}_i(X_\star) + s_i, \quad 1 \le i \le p, \quad (7)$$

where $\mathcal{A}_i$ is the measurement operator,

$$\mathcal{A}_i(X_\star) = \frac{1}{p}\langle A_i X_\star \rangle,$$

$A_i \in \mathbb{R}^{m \times n}$ is the $i$-th measurement matrix, and $s_i$'s correspond to arbitrary and sparse corruptions.

The observation model above can be written in a more compact form as

$$\mathbf{y} = \mathcal{A}(X_\star) + \mathbf{s}. \quad (8)$$

Our goal is to find $X_\star$ given $\mathbf{y}$ and the measurement ensemble $\mathcal{A}(\cdot) = \{\mathcal{A}_i(\cdot)\}_{i=1}^p$. We formulate the problem as

$$\min_{X \in \mathbb{R}^{m \times n}} \|\mathbf{y} - \mathcal{A}(X)\|_1, \quad (9)$$

where we have used the $\ell_1$ norm as the loss term $l(\cdot)$, also known as the residual sum of absolute errors, which is known to be robust to the presence of arbitrary sparse corruptions (Candès et al., 2011).

Recall that we assume we do not know the true rank $r$ of the unknown matrix $X_\star$, and we also solve the problem in the space of matrix factors $L \in \mathbb{R}^{m \times d}$ and $R \in \mathbb{R}^{n \times d}$ whose product equals $X$ i.e., $X = LR^\top$ with $d$ being an overestimate of $r$. We thus formulate matrix sensing as

$$\min_{L \in \mathbb{R}^{m \times d}, R \in \mathbb{R}^{n \times d}} \|\mathbf{y} - \mathcal{A}(LR^\top)\|_1. \quad (10)$$

## 4. Overparameterized Preconditioned Subgradient Algorithm

To minimize the objective function given in (6), we use quasi-Newton type updates. Hence, we use local upper-bounds of the objective function, which lead to preconditioned-type updates for the matrix factors $L$ and $R$, i.e.,

$$(L_{t+1}, R_{t+1}) \equiv \arg\min_{L, R} \mathcal{L}(L_t R_t^\top) + \langle \partial_L \mathcal{L}(L_t R_t^\top), L - L_t \rangle$$
$$+ \langle \partial_R \mathcal{L}(L_t R_t^\top), (R - R_t) \rangle$$
$$+ \frac{1}{2\eta_t}\Big(\|(L - L_t)(R_t^\top R_t + \lambda I)\|_F^2$$
$$+ \|(R - R_t)(L_t^\top L_t + \lambda I)\|_F^2\Big),$$

where $\eta_t$ is the step size and $\partial_L \mathcal{L}(L_t R_t^\top)$ and $\partial_R \mathcal{L}(L_t R_t^\top)$ denote subgradients of the objective function $\mathcal{L}(L_t R_t^\top)$ w.r.t. $L$ and $R$, respectively.

Note that the RHS of the above optimization problem corresponds to upper bounds of the original objective and leads to quasi-Newton-type updates, (Giampouras et al., 2020). We use a Polyak's type step size, and get a similar form to the one in (Tong et al., 2021b), i.e.,

$$\eta_t = \frac{\mathcal{L}(L_t R_t^\top) - \mathcal{L}(L_\star R_\star^\top)}{\gamma_t}, \quad (11)$$

where

$$\gamma_t = \|S_t R_t (R_t^\top R_t + \lambda I)^{-\frac{1}{2}}\|_F^2$$
$$+ \|S_t^\top L_t (L_t^\top L_t + \lambda I)^{-\frac{1}{2}}\|_F^2$$

with $S_t$ denoting a subgradient of the objective $\mathcal{L}(X_t)$, i.e., $S_t \in \partial_X \mathcal{L}(X_t)$.

The proposed algorithm is given in Algorithm 1.

---

**Algorithm 1** Overparameterized Preconditioned Subgradient Algorithm (OPSA)

1: **Input:** Data ($\mathbf{y} \in \mathbb{R}^p$ in the matrix sensing case), $d$: overestimated rank, $\eta$: stepsize.
2: **Initialize** $L_0$ and $R_0$, set $t = 0$.
3: **while** ! Stop Condition **do**
4:      $L_{t+1} = L_t - \eta_t S_t R_t \left(R_t^\top R_t + \lambda I\right)^{-1}$
5:      $R_{t+1} = R_t - \eta_t S_t^\top L_t \left(L_t^\top L_t + \lambda I\right)^{-1}$
6:      $t = t + 1$
7: **end while**
8: **Output:** $\hat{X} = L_t R_t^\top$.

---

*Remark* 4.1. In practice, if $\mathcal{L}(L_\star R_\star^\top)$ is unknown, e.g., due to noise or additional regularization terms, Polyak's type step size may be hard to apply. To address that, one can use a geometrically decaying step size schedule to match the expected linear convergence. Such a step size schedule was introduced in (Goffin, 1977) and has been widely used in the literature on subgradient methods.

## 5. Convergence Analysis

In this section, we present the convergence analysis of the proposed Overaparametrized Preconditioned Subgradient Algorithm (OPSA).

### 5.1. Landscape Assumptions

In the following, we provide the assumptions used in our theoretical results.

**Assumption 5.1** (Restricted Lipschitz Continuity)**.** A function $\mathcal{L}(\cdot) : \mathbb{R}^{m \times n} \mapsto \mathbb{R}$ is rank-$d$ restricted $L$-Lipschitz

continuous for some quantity $L > 0$ if

$$|\mathcal{L}(\boldsymbol{X}_1) - \mathcal{L}(\boldsymbol{X}_2)| \leq L\|\boldsymbol{X}_1 - \boldsymbol{X}_2\|_F \qquad (12)$$

holds for any $\boldsymbol{X}_1, \boldsymbol{X}_2 \in \mathbb{R}^{m \times n}$ such that $\boldsymbol{X}_1 - \boldsymbol{X}_2$ has rank at most $2d$.

**Assumption 5.2** (Restricted Sharpness). A function $\mathcal{L}(\cdot) : \mathbb{R}^{m \times n} \mapsto \mathbb{R}$ is rank-$d$ restricted $\mu$-sharp w.r.t. $\boldsymbol{X}_\star$ for some $\mu > 0$ if

$$\mathcal{L}(\boldsymbol{X}) - \mathcal{L}(\boldsymbol{X}_\star) \geq \mu\|\boldsymbol{X} - \boldsymbol{X}_\star\|_F \qquad (13)$$

holds for any $\boldsymbol{X} \in \mathbb{R}^{m \times n}$ with rank at most $d$.

## 5.2. Main Results

Let $\boldsymbol{F}_\star = \begin{bmatrix} \boldsymbol{L}_\star \\ \boldsymbol{R}_\star \end{bmatrix} \in \mathbb{R}^{(m+n) \times d}$, where $\boldsymbol{L}_\star = \boldsymbol{U}_\star \boldsymbol{\Sigma}_\star^{\frac{1}{2}}$, $\boldsymbol{R}_\star = \boldsymbol{V}_\star \boldsymbol{\Sigma}_\star^{\frac{1}{2}}$, and $\boldsymbol{F} = \begin{bmatrix} \boldsymbol{L} \\ \boldsymbol{R} \end{bmatrix} \in \mathbb{R}^{(m+n) \times d}$.

In our results, we propose a modified version of the distance metric introduced in (Tong et al., 2021b), defined as

$$\text{dist}^2(\boldsymbol{F}, \boldsymbol{F}_\star) = \inf_{\boldsymbol{Q} \in G(d)} \|(\boldsymbol{L}\boldsymbol{Q} - \boldsymbol{L}_\star)(\boldsymbol{\Sigma}_\star + \lambda \boldsymbol{I})^{\frac{1}{2}}\|_F^2$$
$$+ \|(\boldsymbol{R}\boldsymbol{Q}^{-\top} - \boldsymbol{R}_\star)(\boldsymbol{\Sigma}_\star + \lambda \boldsymbol{I})^{\frac{1}{2}}\|_F^2.$$

Matrix $\boldsymbol{Q}$ is known as the alignment matrix and has long been used in similar distance metrics, (Tong et al., 2021a).

**Lemma 5.3.** *The proposed distance metric is an upper bound of the distance $\|\boldsymbol{L}\boldsymbol{R}^\top - \boldsymbol{X}_\star\|_F$. Namely, assuming that $\text{dist}(\boldsymbol{F}, \boldsymbol{F}_\star) \leq \lambda\epsilon$, it holds*

$$\|\boldsymbol{L}\boldsymbol{R}^\top - \boldsymbol{X}_\star\|_F \leq \left(1 + \frac{\epsilon}{2}\right)\sqrt{2}\text{dist}(\boldsymbol{F}, \boldsymbol{F}_\star). \qquad (14)$$

*Proof:* The proof is deferred to supplementary material.

We next provide our main result, which relies on the assumption that the Lipschitz continuity and restricted sharpness condition are satisfied and ensure exact convergence.

**Theorem 5.4** (Convergence of OPSA). *Let $\mathcal{L}(\boldsymbol{X}) : \mathbb{R}^{m \times n} \mapsto \mathbb{R}$ be convex w.r.t $\boldsymbol{X}$, and assume that it satisfies the rank-$d$ restricted $L$-Lipschitz continuity assumption and the rank-$d$ restricted $\mu$-sharpness condition, defined above. Let also $\lambda = \frac{1}{20}$, and without loss of generality $\sigma_r(\boldsymbol{X}_\star) = 1$. Let $\epsilon = \frac{10^{-4}}{\chi\|\boldsymbol{X}_\star\|_{op}^{\frac{1}{2}}}$, and*

$$\text{dist}(\boldsymbol{F}_0, \boldsymbol{F}_\star) \leq \lambda\epsilon = \frac{10^{-4}}{\chi}\frac{\|\boldsymbol{X}_\star\|_{op}^{-\frac{1}{2}}}{20}, \qquad (15)$$

*where $\chi = \frac{L}{\mu}$. Then for the Overparameterized Preconditioned Subgradient Algorithm (OPSA) given in Algorithm 1,*

*with the Polyak's step-size defined in* (11), *we have*

$$\text{dist}(\boldsymbol{F}_t, \boldsymbol{F}_\star) \leq \left(1 - \frac{0.12}{\chi^2}\right)^{\frac{t}{2}} \frac{10^{-4}\|\boldsymbol{X}_\star\|_{op}^{-\frac{1}{2}}}{20 \cdot \chi},$$

$$\|\boldsymbol{L}_t\boldsymbol{R}_t^\top - \boldsymbol{X}_\star\|_F \leq \left(1 - \frac{0.12}{\chi^2}\right)^{\frac{t}{2}} \frac{1.5 \times 10^{-4}\|\boldsymbol{X}_\star\|_{op}^{-\frac{1}{2}}}{20 \cdot \chi}.$$

*Proof.* The rate can be derived from its general form (see (16)) by setting $\lambda = \frac{\|\boldsymbol{X}_\star\|_{op}}{\bar{\lambda}}$, $\bar{\lambda} = c\|\boldsymbol{X}_\star\|_{op}$, $c = 20$. The detailed proof is deferred to supplementary material. $\square$

*Remark* 5.5. Our derived rate of convergence requires a good initialization that will satisfy condition (15). In practice, this condition can be satisfied by using a truncated spectral method as in (Zhang et al., 2016). It should be also noted that, even though the rate is independent of the condition number of $\boldsymbol{X}_\star$, the initialization condition is negatively affected as this condition number increases, requiring initializations closer to $\boldsymbol{X}_\star$.

**Technical innovation.** It should be noted that the derivation of the convergence rate in the *overpararameterized and asymmetric* case that we focus on, requires a novel definition of the distance metric given in (23). Namely, previous convergence metrics dealing with the imbalance issue that shows up in the asymmetric low-rank matrix estimation problems, cannot be applied in the overparameterized setting due to overparameterization and non-invertibility of $\boldsymbol{\Sigma}_\star$. In order to prove the contraction of this convergence metric we derive novel and non-trivial perturbation-type bounds of matrix norms, which pose their own challenges and move beyond existing approaches that address either the known-rank or unknown-rank and symmetric matrices regime (see Table 1).

Note also that the derived rate of convergence is valid in the overparameterized rank setting where $d \geq r$ and the value of $\lambda$ should be positive i.e., $\lambda > 0$. The positive value of $\lambda$ is required so that the matrix $\boldsymbol{\Sigma}_\star + \lambda I$ that is used in the convergence metric is always invertible. When $d = r$ the rate of OPSA can take the same form as the one of the scaled subgradient method of (Tong et al., 2021b), that is derived in the known rank regime. Namely, when $d = r$ then $\boldsymbol{\Sigma}_\star$ is invertible and $\lambda$ could be 0. To make the theorem applicable for $d = r$, we could replace the $\lambda$ parameter appearing in (15), with $\lambda' = \lambda + \sigma_d(\boldsymbol{X}_\star)$. In that case, when $d > r$, we have $\lambda' = \lambda$ since $\sigma_d(\boldsymbol{X}_\star) = 0$. But when $d = r, \lambda' = \sigma_r(\boldsymbol{X}_\star)$ (since $\lambda = 0$),and the initialization condition and rate of convergence (see supplementary material) boils down to a similar form, up to some constants, to the one in (Tong et al., 2021b).

*Remark* 5.6. As is shown in Theorem 5.4, the rate is independent of the condition number of $\boldsymbol{X}_\star$. The linear rate provided in Theorem 5.4 is a simplified form of the general

expression:

$$\rho(\chi, \epsilon, \lambda) = 1$$

$$- \frac{1}{2\chi^2} \sqrt{\frac{\sqrt{2}-1}{1+2\lambda}} \left( \sqrt{\frac{\sqrt{2}-1}{1+2\lambda}} \left( 2 - \frac{1}{(1-\sqrt{\epsilon}(\sqrt{\epsilon}+\sqrt{2}\bar{\lambda}^{\frac{1}{4}}))^2} \right) \right.$$

$$\left. - 2\chi\sqrt{2} \left( \frac{3}{2}\epsilon + 2\frac{\sqrt{\epsilon}(\sqrt{\epsilon}+\sqrt{2}\bar{\lambda}^{\frac{1}{4}})}{1-\sqrt{\epsilon}(\sqrt{\epsilon}+\sqrt{2}\bar{\lambda}^{\frac{1}{4}})} \right) \right).$$

$$(16)$$

Note that this general form holds for any value of $\lambda \geq 0$, and its detailed derivation can be found in the supplementary material (Theorem A.3.1).

*Remark* 5.7. The level of overparameterization influences the rate of convergence through $\chi$. Specifically, $\chi$ depends on the landscape properties of the objective as expressed by the Lipschitcz constant $L$ and the restricted sharpness constant $\mu$. Generally speaking, $\chi = \frac{L}{\mu}$ increases with $d$ ($L \uparrow, \mu \downarrow$) making the rate of convergence slower. This effect of the overestimated rank $d$ on the rate of convergence is demonstrated in the numerical experiments section (see Figure 2).

## 5.3. Iteration Complexity for Matrix Sensing

In this section, we analyze the iteration complexity, i.e., the number of iterates $T$ required to reach $\|\boldsymbol{L}_T \boldsymbol{R}_T^\top - \boldsymbol{X}_\star\|_F \leq \epsilon$, namely $\epsilon$-accuracy. This study will focus on the matrix sensing problem.

### 5.3.1. NOISELESS MATRIX SENSING

Herein, we focus on the noiseless case without the presence of outliers, where it holds that $\mathbf{y} = \mathcal{A}(\boldsymbol{X}_\star)$.

**Definition 5.8** (Mixed-norm RIP). Let $2d > 0$ denote the rank of $\boldsymbol{X}$ and $\mathcal{A}(\cdot)$ a linear map. We define the $\delta_{2d}^-, \delta_{2d}^+$, as the lower and upper uniform bounds, respectively, of the quantity $\frac{\|\mathcal{A}(\boldsymbol{X})\|_1}{\|\boldsymbol{X}\|_F}$ for all matrices $\boldsymbol{X}$ of rank at most $2d$.

The mixed-norm RIP is empirically verified for the Gaussian linear map in Figure 1.

**Proposition 5.9** (Lipschitz Continuity and Restricted Sharpness–No Outliers). *If $\mathcal{A}(\cdot)$ satisfies the mixed-norm property with constants $\delta_{2d}^-, \delta_{2d}^+$, then $\mathcal{L}(\boldsymbol{X}) = \|\mathbf{y} - \mathcal{A}(\boldsymbol{X})\|_1$ satisfies the rank-$d$ restricted $L$-Lipschitz continuity and rank-$d$ restricted sharpness with constants,*

$$L = \delta_{2d}^+ \quad and \quad \mu = \delta_{2d}^-. \quad (17)$$

*Proof:* Similar to Proposition 1 in (Tong et al., 2021b).

**Corollary 5.10** (Iteration Complexity–No Outliers). *Using the same setting as in Theorem 5.4, we use the Lipschitz constant and restricted sharpness constant of Proposition 5.9, we get iteration complexity for noiseless matrix sensing is $\mathcal{O}\left( \left( \frac{\delta_{2d}^+}{\delta_{2d}^-} \right)^2 \log(\frac{1}{\epsilon}) \right)$.*

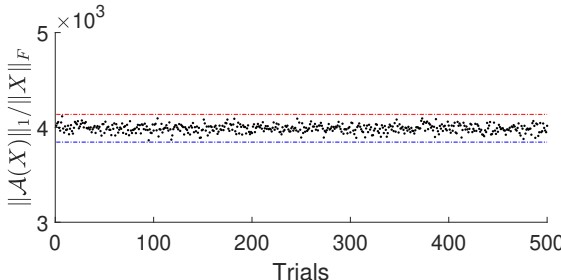

*Figure 1.* Values of $\frac{\|\mathcal{A}(\boldsymbol{X})\|_1}{\|\boldsymbol{X}\|_F}$ for Gaussian map $\mathcal{A} : \mathbb{R}^{500 \times 500} \to \mathbb{R}^{5000}$, where $\mathrm{rank}(\boldsymbol{X}) = 10$. Each point represents one result of 500 random trials. Blue and red dash lines are the lower and upper uniform bounds, respectively.

### 5.3.2. ROBUST MATRIX SENSING

Here we consider the presence of outliers. We thus first define the $\mathcal{S}$-outlier type bound condition. The S-outlier bound property has been used in the robust low-rank matrix recovery problem in prior works, e.g., (Tong et al., 2021b). It actually encodes a property of that allows restricted sharpness condition to be satisfied in matrix sensing problems in the presence of outliers. A detailed derivation of this condition, as a natural generalization of RIP can be found in (Charisopoulos et al., 2021).

**Definition 5.11** ($\mathcal{S}$-outlier Type Bound). The linear map $\mathcal{A}(\cdot)$ satisfies the rank-$2d$ $\mathcal{S}$-outlier type bound w.r.t. a set $\mathcal{S}$ with a constant $\delta^0$ if for all matrices $\boldsymbol{X} \in \mathbb{R}^{m \times n}$ of rank at most $2d$ we have

$$\delta^0 \|\boldsymbol{X}\|_F \leq \|\mathcal{A}_{\mathcal{S}^c}(\boldsymbol{X})\|_1 - \|\mathcal{A}_{\mathcal{S}}(\boldsymbol{X})\|_1, \quad (18)$$

where $\mathcal{A}_{\mathcal{S}}(\boldsymbol{X}) = \{\mathcal{A}_i(\boldsymbol{X})\}_{i \in \mathcal{S}}$ and $\mathcal{A}_{\mathcal{S}^c}(\boldsymbol{X}) = \{\mathcal{A}_i(\boldsymbol{X})\}_{i \in \mathcal{S}^c}$.

**Proposition 5.12** (Matrix Sensing with Outliers). *Let $\mathcal{A}(\cdot)$ the rank-$2d$ mixed-norm RIP with $(\delta_{2d}^-, \delta_{2d}^+)$ and the $\mathcal{S}$-outlier bound property defined above with $\delta^o$. Then $\mathcal{L}(\boldsymbol{X})$ satisfies the rank-$d$ restricted $L$-Lipschitz continuity and $\mu$-sharpness with*

$$L = \delta_{2d}^+ \quad and \quad \mu = \delta^0. \quad (19)$$

*Proof:* The Lipschitz constant can be derived following similar steps as in the noiseless case. For the restricted sharpness constant $\mu$ we use Proposition 2 of (Tong et al., 2021b).

**Corollary 5.13** (Iteration Complexity with Outliers). *Under the same setting as in Theorem 5.4, and by using the Lipschitz constant and restricted sharpness constant of Propositions 5.12, we get iteration complexity for matrix sensing, in the presence of outliers, is $\mathcal{O}\left( \left( \frac{\delta_{2d}^+}{\delta^0} \right)^2 \log(\frac{1}{\epsilon}) \right)$.*

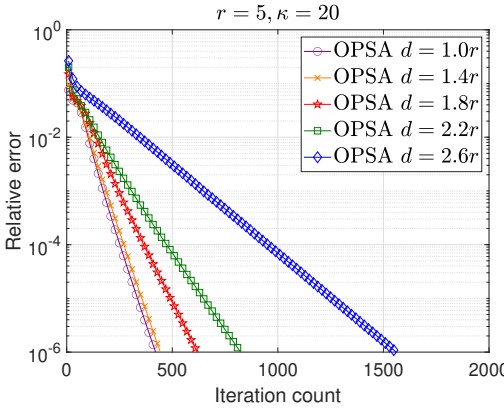
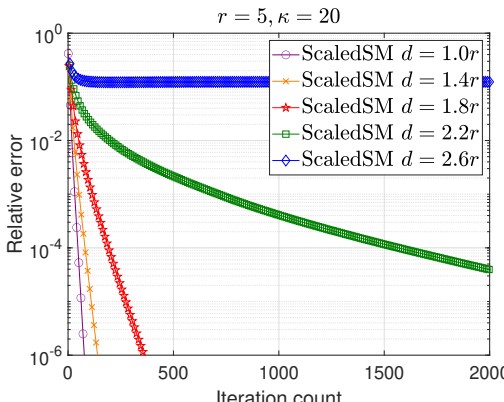

*Figure 2.* Performance comparison between OPSA (top) and ScaledSM (bottom) with different overparameterization $d$, where $n, r, \kappa, \lambda$, outlier $= 100, 5, 20, 2, 10\%$.

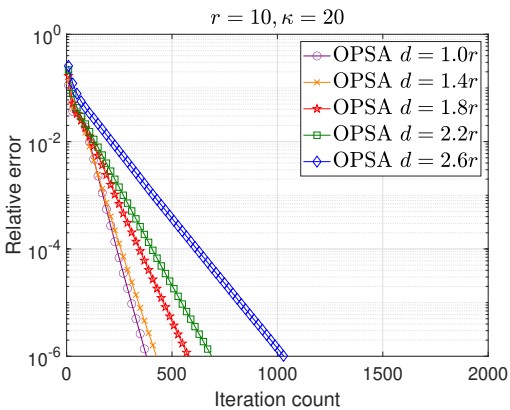
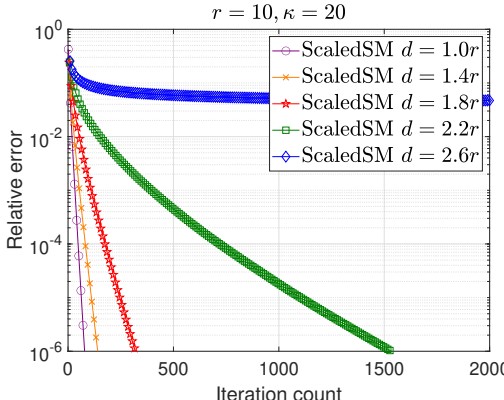

*Figure 3.* Performance comparison between OPSA (top) and ScaledSM (bottom) with different overparameterization $d$, where $n, r, \kappa, \lambda$, outlier $= 100, 10, 20, 2, 10\%$.

It should be noted that Corollary 5.13 implies that in the presence of outliers, the iteration complexity increases at an amount depending on the properties of the measurement matrices (through $\delta^0$).

*Remark* 5.14. Assuming a measurement operator $\mathcal{A}_i(\boldsymbol{X}) = \frac{1}{p}\langle \boldsymbol{A}_i, \boldsymbol{X} \rangle$, where $\boldsymbol{A}_i$ has Gaussian i.i.d. entries $\mathcal{N}(0, 1)$, we can invoke the results of (Charisopoulos et al., 2021; Tong et al., 2021b), which show that $\mathcal{A}(\cdot)$ satisfies the mixed-RIP and and $\mathcal{S}$-outlier bound conditions with

$$\delta_{2d}^- \gtrsim 1, \quad \delta_{2d}^+ \lesssim 1, \quad \delta_0 \gtrsim 1 - 2p_s,$$

where $p_s \in [0, \frac{1}{2})$ is the fraction of outliers, as long as

$$p \gtrsim \frac{(m+n)d}{(1-2p_s)^2} \log\left(\frac{1}{1-2p_s}\right).$$

Hence, under the same setting as in Theorem 5.4 OPSA converges linearly to $\epsilon$-accuracy in

$$O\left(\frac{1}{(1-2p_s)^2} \log\frac{1}{\epsilon}\right)$$

iterations assuming that it is initialized appropriately (see statement of Theorem 5.4).

# 6. Numerical Experiments

In this section, we verify the empirical performance of the proposed overparameterized Preconditioned Subgradient Algorithm (OPSA), i.e., Algorithm 1.

**Experimental setup.** The ground truth $\boldsymbol{X}_\star$ is generated as a product of two $n \times r$ random matrices, and then the condition number is adjusted to be $\kappa$ by altering the singular values of $\boldsymbol{X}_\star$. The observations of matrix sensing are obtained as described in (8), that is

$$\mathbf{y}_i = \mathcal{A}_i(\boldsymbol{X}_\star) + \mathbf{s}_i, \qquad 1 \le i \le p,$$

where the sensing operator $\mathcal{A}_i(\boldsymbol{X}_\star) = \frac{1}{p}\langle \boldsymbol{A}_i, \boldsymbol{X}_\star \rangle$ and $\boldsymbol{A}_i$ is a Gaussian random matrix. The corruption vector $\mathbf{s} \in \mathbb{R}^p$ contains randomly positioned outliers whose values are drawn uniformly at random from the interval $[-10\|\mathcal{A}(\boldsymbol{X}_\star)\|_\infty, 10\|\mathcal{A}_i(\boldsymbol{X}_\star)\|_\infty]$. Through this section,

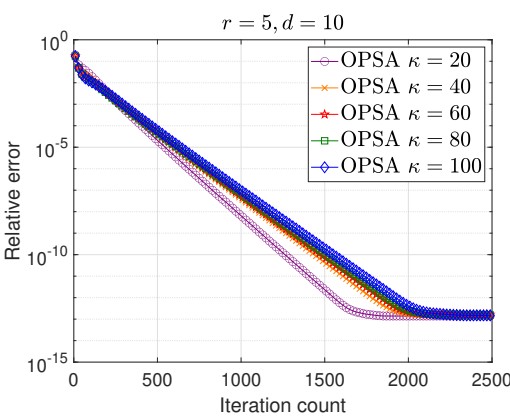 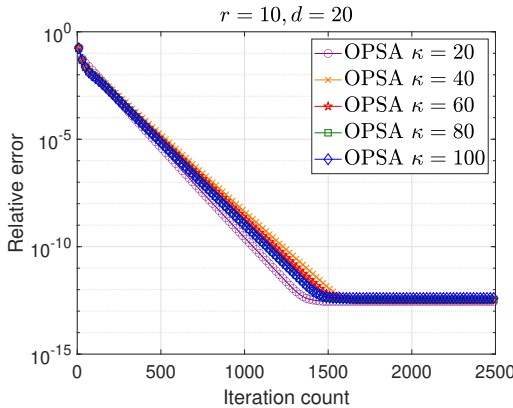

*Figure 4.* OPSA performance with different condition numbers $\kappa$, where $n, \lambda,$ outlier $= 100, 2, 10\%$. **Top**: $r, d = 5, 10$. **Bottom**: $r, d = 10, 20$.

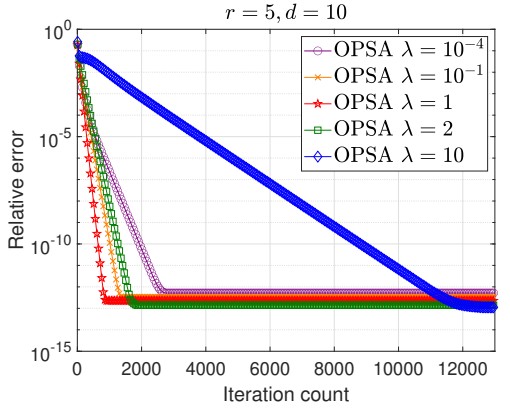 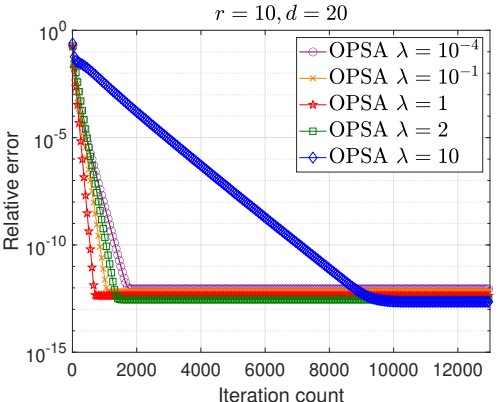

*Figure 5.* OPSA performance with different $\lambda$, where $n, \kappa,$ outlier $= 100, 20, 10\%$. **Top**: $r, d = 5, 10$. **Bottom**: $r, d = 10, 20$.

$m = 8nr$ measurements are used in all tests. The code of ScaledSM is obtained from the authors' website. For the sake of fairness, Polyak's stepsizes are used for all tested algorithms, thus ScaledSM requires no extra parameter tuning. For OPSA, $\lambda$ is the only parameter to be tuned, and an experiment in this section shows that $\lambda$ can be easily tuned in a wide favorable range. All the tests are conducted with Matlab 2024a on a mobile workstation equipped with an Intel i9-12950HX CPU and 64GB of RAM. The Matlab implementation for the proposed OPSA is available online at https://github.com/caesarcai/OPSA.

**Performance with different overparameterizations $d$.** We compare the convergence performance of OPSA against ScaledSM (Tong et al., 2021b), the state-of-the-art subgradient method for robust matrix sensing, with different levels of overparameterization $d$. The convergence is evaluated with respect to the relative error $\|L_t R_t^\top - X_\star\|_F / \|X_\star\|_F$. The comparison results with true rank $r = 5$ and $r = 10$ are reported in Figures 2 and 3, respectively. For both algorithms, one can see that more overparameterization leads to

slower convergence. However, OPSA constantly achieves linear convergence while ScaledSM fails to converge when major overparameterization happens.

**Performance with different condition numbers $\kappa$.** In this experiment, we test the convergence performance of OPSA with different condition numbers of $X_\star$. The results with conditional numbers up to 100 are reported in Figure 4. One can observe that the convergence behavior and final accuracy of OPSA are not distinctly affected by larger condition numbers, even when the rank is heavily overestimated. This observation matches our main theoretical result. In Figure 7, we conduct additional experiments for OPSA with extremely large condition numbers $\kappa$ up to $10,000$. The results are consistence with Figure 4. Even with $\kappa = 1,000$ or $\kappa = 10,000$, OPSA still delivers stable convergence and achieves the same final accuracy.

**Performance with different parameters $\lambda$.** Notice that $\lambda$ is the only parameter to be tuned in OPSA. In Figure 5, we test the convergence performance of OPSA with different $\lambda$. One can observe that a mild $\lambda$ value (e.g., 0.1, 1, 2) helps

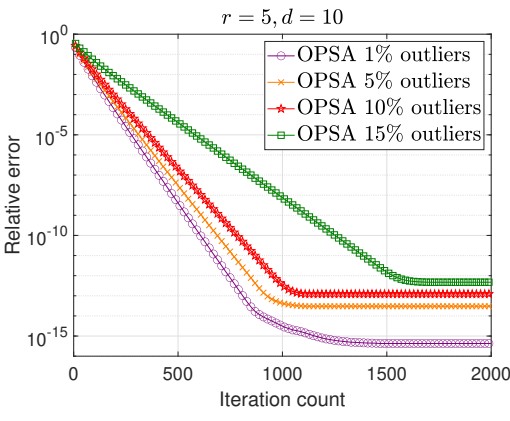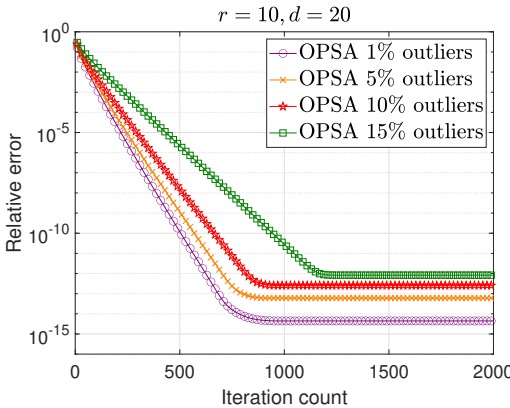

*Figure 6.* OPSA performance with different outlier densities, where $n, \kappa, \lambda = 100, 20, 2$. **Top**: $r, d = 5, 10$. **Bottom**: $r, d = 10, 20$.

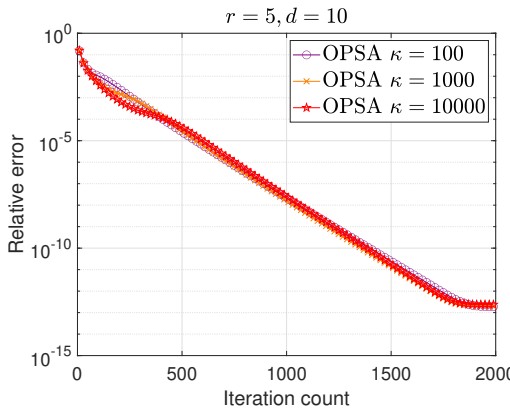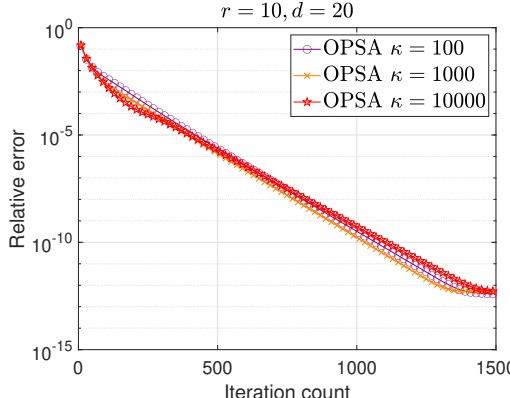

*Figure 7.* OPSA performance with different large condition numbers $\kappa$, where $n, \lambda, \text{outlier} = 100, 2, 10\%$. **Left**: $r, d = 5, 10$. **Right**: $r, d = 10, 20$.

OPSA maintain steep linear convergence when the rank is heavily overestimated. However, if the $\lambda$ parameter is set too large (e.g., 10) or too tiny (e.g., $10^{-4}$), OPSA converges much slower, although still at a linear rate. Overall, OPSA's performance is favorable with a wide range of $\lambda$ values, and thus the parameter tuning is easy for the proposed OPSA.

**Performance with different outlier densities.** In Figure 6, OPSA is tested against different outlier densities. OPSA successfully recovers $X_\star$ and maintains linear convergence in all tests. Note that more iterations are needed, and slightly worse final accuracy can be achieved when more outliers are present. This is as expected since more outliers lead to harder recovery problems.

**More numerical results.** Additional numerical experiments on partially observed video background subtraction tasks, have been conducte and reported in the supplementary material.

## 7. Conclusions

In this paper, we proposed an overparameterized Preconditioned Subgradient Algorithm (OPSA) for robust low-rank matrix recovery. Our work goes beyond existing SOTA works by addressing the challenging scenario of robust low-rank matrix recovery in the case of *asymmetric* matrices of *unknown rank* using a preconditioned-type approach. Under certain landscape assumptions i.e., Lipschitz continuity and restricted sharpness conditions, we a) established a linear rate of convergence that is independent of the condition number of the unknown matrix, and b) derived the iteration complexity matrix sensing, in the noiseless setting and in the presence of outliers. Numerical results corroborate our theoretical findings for different levels of overparameterization of the rank, and outliers, and the independence of the rate of convergence on the condition number of the sought matrix.

## Acknowledgements

H.Q. Cai acknowledges partial support from the National Science Foundation (NSF) through grant DMS-2304489. R. Vidal acknowledges partial support of NSF grants 2212457 and 2031985, and Simons Foundation grant 814201.

## Impact Statement

Low-rank matrix recovery has been broadly applied as one of the fundamental techniques in data mining and machine learning research. Hence, through its applications, the proposed preconditioned subgradient algorithm has potential broader impacts that data mining and machine learning have, as well as their limitations.

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

# Supplementary Materials for
# Guarantees of a Preconditioned Subgradient Algorithm for
# Overparameterized Asymmetric Low-rank Matrix Recovery

## A.1. Problem Formulation

We focus on the following low-rank matrix estimation problem

$$\min_{\boldsymbol{X}} \ \mathcal{L}(\boldsymbol{Y}, \boldsymbol{X}) \equiv \min_{\boldsymbol{L}, \boldsymbol{R}, \boldsymbol{L}\boldsymbol{R}^\top = \boldsymbol{X}} \ \mathcal{L}(\boldsymbol{Y}, \boldsymbol{L}\boldsymbol{R}^\top). \tag{20}$$

We assume a preconditioned alternating gradient descent algorithm consisting of the following steps:

$$\boldsymbol{L}_{t+1} = \boldsymbol{L}_t - \eta_t (\boldsymbol{S}_t \boldsymbol{R}_t) \left( \boldsymbol{R}_t^\top \boldsymbol{R}_t + \lambda \boldsymbol{I} \right)^{-1}, \tag{21}$$

$$\boldsymbol{R}_{t+1} = \boldsymbol{R}_t - \eta_t (\boldsymbol{S}_t^\top \boldsymbol{L}_t) \left( \boldsymbol{L}_t^\top \boldsymbol{L}_t + \lambda \boldsymbol{I} \right)^{-1}, \tag{22}$$

with $\boldsymbol{S}_t \in \partial_{\boldsymbol{X}} l(\boldsymbol{Y}, \boldsymbol{X}_t)$.

Let $\boldsymbol{X}_\star$ be the ground truth matrix, $\boldsymbol{F}_\star = \begin{bmatrix} \boldsymbol{L}_\star \\ \boldsymbol{R}_\star \end{bmatrix} \in \mathbb{R}^{(m+n) \times d}$, where $d \geq r = \text{rank}(\boldsymbol{X}_\star)$, $\boldsymbol{L}_\star = \boldsymbol{U}_\star \boldsymbol{\Sigma}_\star^{\frac{1}{2}}$, $\boldsymbol{R}_\star = \boldsymbol{V}_\star \boldsymbol{\Sigma}_\star^{\frac{1}{2}}$, and

$\boldsymbol{F} = \begin{bmatrix} \boldsymbol{L} \\ \boldsymbol{R} \end{bmatrix} \in \mathbb{R}^{(m+n) \times d}$. We define the following convergence metric,

$$\text{dist}(\boldsymbol{F}, \boldsymbol{F}_\star) = \sqrt{\inf_{\boldsymbol{Q} \in \text{G}(d)} \|(\boldsymbol{L}\boldsymbol{Q} - \boldsymbol{L}_\star)(\boldsymbol{\Sigma}_\star + \lambda \boldsymbol{I})^{\frac{1}{2}}\|_F^2 + \|(\boldsymbol{R}\boldsymbol{Q}^{-\top} - \boldsymbol{R}_\star)(\boldsymbol{\Sigma}_\star + \lambda \boldsymbol{I})^{\frac{1}{2}}\|_F^2}. \tag{23}$$

In our analysis we focus on the Polyak's type step size, defined as follows,

$$\eta_t = \frac{\mathcal{L}(\boldsymbol{L}_t \boldsymbol{R}_t^\top) - \mathcal{L}(\boldsymbol{L}_\star \boldsymbol{R}_\star^\top)}{\|\boldsymbol{S}_t \boldsymbol{R}_t (\boldsymbol{R}_t^\top \boldsymbol{R}_t + \lambda \boldsymbol{I})^{-\frac{1}{2}}\|_F^2 + \|\boldsymbol{S}_t^\top \boldsymbol{L}_t (\boldsymbol{L}_t^\top \boldsymbol{L}_t + \lambda \boldsymbol{I})^{-\frac{1}{2}}\|_F^2}. \tag{24}$$

---

**Algorithm 1** Overparameterized Preconditioned Subgradient Algorithm (OPSA)

---

1: **Input:** $\mathbf{y} \in \mathbb{R}^p$, $d$: overestimated rank, $\lambda > 0$: regularization parameter, $\eta$: stepsize.
2: **Initialize** $\boldsymbol{L}_0$ and $\boldsymbol{R}_0$, set $t = 0$.
3: **while** ! Stop Condition **do**
4:      $\boldsymbol{L}_{t+1} = \boldsymbol{L}_t - \eta_t \boldsymbol{S}_t \boldsymbol{R}_t \left( \boldsymbol{R}_t^\top \boldsymbol{R}_t + \lambda \boldsymbol{I} \right)^{-1}$
5:      $\boldsymbol{R}_{t+1} = \boldsymbol{R}_t - \eta_t \boldsymbol{S}_t^\top \boldsymbol{L}_t \left( \boldsymbol{L}_t^\top \boldsymbol{L}_t + \lambda \boldsymbol{I} \right)^{-1}$
6:      $t = t + 1$
7: **end while**
8: **Output:** $\boldsymbol{X} = \boldsymbol{L}_t \boldsymbol{R}_t^\top$.

---

## A.2. Auxiliary Lemmata

**Lemma A.2.1** (Theorem 1 (Birman et al., 1975); Theorem X.1.1, (Bhatia, 2013)). *Let $A$, $B$ positive definite matrices. It holds,*

$$\|A^{\frac{1}{2}} - B^{\frac{1}{2}}\|_{op} \le \|A - B\|_{op}^{\frac{1}{2}}. \tag{25}$$

**Lemma A.2.2.** *Let $\|X - X_\star\|_{op} \le \frac{1}{2}\sigma_r(X_\star)$, then it holds $\|X - X_\star\|_F \ge \sqrt{\frac{(\sqrt{2}-1)\sigma_r(X_\star)}{\sigma_r(X_\star)+2\lambda}}\text{dist}(F, F_\star)$.*

*Proof:* Let us denote $Q_\star$ as the optimal alignment matrix for $\text{dist}(F, F_\star)$. We will have,

$$
\begin{aligned}
\text{dist}^2(F, F_\star) &= \|(LQ_\star - L_\star)(\Sigma_\star + \lambda I)^{\frac{1}{2}}\|_F^2 + \|(RQ_\star^{-\top} - R_\star)(\Sigma_\star + \lambda I)^{\frac{1}{2}}\|_F^2 \\
&= \text{tr}\{(\Sigma_\star + \lambda I)^{\frac{1}{2}}(LQ_\star - L_\star)^\top(LQ_\star - L_\star)(\Sigma_\star + \lambda I)^{\frac{1}{2}}\} \\
&\quad + \text{tr}\{(\Sigma_\star + \lambda I)^{\frac{1}{2}}(RQ_\star^{-\top} - R_\star)^\top(RQ_\star^{-\top} - R_\star)(\Sigma_\star + \lambda I)^{\frac{1}{2}}\} \\
&= \text{tr}\{(LQ_\star - L_\star)^\top(LQ_\star - L_\star)(\Sigma_\star + \lambda I)\} + \text{tr}\{(RQ_\star^{-\top} - L_\star)^\top(RQ_\star^{-\top} - R_\star)(\Sigma_\star + \lambda I)\} \\
&= \|(LQ_\star - L_\star)\Sigma_\star^{\frac{1}{2}}\|_F^2 + \|(RQ_\star^{-\top} - R_\star)\Sigma_\star^{\frac{1}{2}}\|_F^2 + \lambda\left(\|(LQ_\star - L_\star)\|_F^2 + (RQ_\star^{-\top} - R_\star)\|_F^2\right).
\end{aligned}
$$

By using the definition of the distance and Lemma 11 in (Tong et al., 2021a), and Lemma 5.14 in (Tu et al., 2016), we have

$$\text{dist}^2(F, F_\star) \le \frac{1}{\sqrt{2}-1}\|X - X_\star\|_F^2 + \frac{2\lambda}{(\sqrt{2}-1)\sigma_r(X_\star)}\|X - X_\star\|_F^2. \tag{26}$$

Hence, we have

$$\text{dist}^2(F, F_\star) \le \left(\frac{1}{\sqrt{2}-1} + \frac{2\lambda}{(\sqrt{2}-1)\sigma_r(X_\star)}\right)\|X - X_\star\|_F^2 = \frac{\sigma_r(X_\star) + 2\lambda}{(\sqrt{2}-1)\sigma_r(X_\star)}\|X - X_\star\|_F^2, \tag{27}$$

which leads us to inequality

$$\|X - X_\star\|_F \ge \sqrt{\frac{(\sqrt{2}-1)\sigma_r(X_\star)}{\sigma_r(X_\star) + 2\lambda}}\text{dist}(F, F_\star). \tag{28}$$

**Lemma A.2.3.** *The proposed distance metric given in (23), is an upper bound of the distance $\|LR^\top - X_\star\|_F$. Namely, assuming that $\text{dist}(F, F_\star) \le \lambda\epsilon$, it holds*

$$\|LR^\top - X_\star\|_F \le \left(1 + \frac{\epsilon}{2}\right)\sqrt{2}\text{dist}(F, F_\star). \tag{29}$$

*Proof:* We use the following distance metric,

$$\text{dist}(F, F_\star) = \sqrt{\inf_{Q \in \text{G}(d)} \|(LQ - L_\star)(\Sigma_\star + \lambda I)^{\frac{1}{2}}\|_F^2 + \|(RQ^{-\top} - R_\star)(\Sigma_\star + \lambda I)^{\frac{1}{2}}\|_F^2} \tag{30}$$

and we assume that $\text{dist}(F_t, F_\star) \le \lambda\epsilon$.

By using the following known norm inequalities

$$\|AB\|_F \ge \|A\|_F\sigma_r(B) \ge \|A\|_{op}\sigma_r(B), \tag{31}$$

we can get

$$\|(L - L_\star)(\Sigma_\star + \lambda I)^{-\frac{1}{2}}\|_{op}^2\lambda^2 + \|(R - R_\star)(\Sigma_\star + \lambda I)^{-\frac{1}{2}})\|_{op}^2\lambda^2 \le \lambda^2\epsilon^2, \tag{32}$$

which implies

$$\max\{\|(L - L_\star)(\Sigma_\star + \lambda I)^{-\frac{1}{2}}\|_{op}, \|(R - R_\star)(\Sigma_\star + \lambda I)^{-\frac{1}{2}})\|_{op}\} \le \epsilon. \tag{33}$$

Next, we show the relation between $\|L_t R_t^\top - X_\star\|_F$ and $\mathrm{dist}(F_t, F_\star)$,

$$
\begin{aligned}
\|L_t R_t^\top - X_\star\|_F &= \|LR^\top - X_\star\|_F = \|\Delta_L R_\star^\top + L_\star \Delta_R^\top + \Delta_L \Delta_R^\top\|_F \\
&\leq \|\Delta_L R_\star^\top\|_F + \|L_\star \Delta_R^\top\|_F + \|\Delta_L \Delta_R^\top\| = \|\Delta_L R_\star^\top\|_F + \|L_\star \Delta_R^\top\|_F \\
&\quad + \|\frac{1}{2}\Delta_L (\Sigma_\star + \lambda I)^{\frac{1}{2}} (\Sigma_\star + \lambda I)^{-\frac{1}{2}} \Delta_R^\top + \frac{1}{2}\Delta_L (\Sigma_\star + \lambda I)^{\frac{1}{2}} (\Sigma_\star + \lambda I)^{-\frac{1}{2}} \Delta_R^\top\|_F \\
&\leq \|\Delta_L(\Sigma_\star + \lambda I)^{\frac{1}{2}}\|_F + \|\Delta_R(\Sigma_\star + \lambda I)^{\frac{1}{2}} I\|_F + \frac{1}{2}\|\Delta_L (\Sigma_\star + \lambda I)^{\frac{1}{2}}\|_F\|\Delta_R \left(\Sigma_\star + \lambda I\right)^{-\frac{1}{2}}\|_{op} \\
&\quad + \frac{1}{2}\|\Delta_L (\Sigma_\star + \lambda I)^{-\frac{1}{2}}\|_{op}\|\Delta_R \left(\Sigma_\star + \lambda I\right)^{\frac{1}{2}}\|_F \\
&\leq \left(1 + \frac{1}{2}\max\{\|\Delta_R \left(\Sigma_\star + \lambda I\right)^{-\frac{1}{2}}\|_{op}, \|\Delta_L \left(\Sigma_\star + \lambda I\right)^{-\frac{1}{2}}\|_{op}\}\right) \left(\|\Delta_L \left(\Sigma_\star + \lambda I\right)^{\frac{1}{2}}\|_F + \|\Delta_R \left(\Sigma_\star + \lambda I\right)^{\frac{1}{2}}\|_F\right) \\
&\leq \left(1 + \frac{\epsilon}{2}\right) \sqrt{2}\mathrm{dist}\left(F, F_\star\right) \leq \left(1 + \frac{\epsilon}{2}\right) \sqrt{2}\lambda\epsilon,
\end{aligned}
$$

where he have used the initialization condition, the fact that

$$
\max\{\|\Delta_R \left(\Sigma_\star + \lambda I\right)^{-\frac{1}{2}}\|_{op}, \|\Delta_L \left(\Sigma_\star + \lambda I\right)^{-\frac{1}{2}}\|_{op}\} \leq \epsilon \tag{34}
$$

and the inequality

$$
\begin{aligned}
&\|\Delta_L \left(\Sigma_\star + \lambda I\right)^{\frac{1}{2}}\|_F + \|\Delta_R \left(\Sigma_\star + \lambda I\right)^{\frac{1}{2}}\|_F \\
&\leq \sqrt{2}\sqrt{\|\Delta_L \left(\Sigma_\star + \lambda I\right)^{\frac{1}{2}}\|_F^2 + \|\Delta_R \left(\Sigma_\star + \lambda I\right)^{\frac{1}{2}}\|_F^2}.
\end{aligned} \tag{35}
$$

This finishes the proof. $\qquad\square$

**Lemma A.2.4.** *Let us assume* $\mathrm{dist}(F, F_\star) \leq \epsilon\lambda$, *where* $\lambda = \frac{\|X_\star\|_{op}}{\bar\lambda}$. *Let* $\sigma_r(X_\star) = 1$ *and denote* $\Delta_\lambda R = (R^\top R + \lambda I)^{1/2} - (R_\star^\top R_\star + \lambda I)^{1/2}$, $\Delta_\lambda L = (L^\top L + \lambda I)^{1/2} - (L_\star^\top L_\star + \lambda I)^{1/2}$, *then it holds*

$$
\begin{aligned}
\|(\Sigma_\star + \lambda I)^{-\frac{1}{2}} \Delta_\lambda L\|_{op} &\leq \sqrt{\epsilon}(\sqrt{\epsilon} + \sqrt{2}\bar\lambda^{\frac{1}{4}}), \\
\|(\Sigma_\star + \lambda I)^{-\frac{1}{2}} \Delta_\lambda R\|_{op} &\leq \sqrt{\epsilon}(\sqrt{\epsilon} + \sqrt{2}\bar\lambda^{\frac{1}{4}}).
\end{aligned} \tag{36}
$$

*Proof:* We have

$$
\|(\Sigma_\star + \lambda I)^{-\frac{1}{2}} \Delta_\lambda L\|_{op} = \|\left((L^\top L + \lambda I)(\Sigma_\star + \lambda I)^{-1}\right)^{\frac{1}{2}} - \left((L_\star^\top L_\star + \lambda I)(\Sigma_\star + \lambda I)^{-1}\right)^{\frac{1}{2}}\|_{op}. \tag{37}
$$

Let us denote $A = (L^\top L + \lambda I)(\Sigma_\star + \lambda I)^{-1}$ and $B = (L_\star^\top L_\star + \lambda I)(\Sigma_\star + \lambda I)^{-1}$. From Lemma A.2.1, we have that $\|A^{\frac{1}{2}} - B^{\frac{1}{2}}\|_{op} \leq \|\|A - B\|\|_{op}^{\frac{1}{2}}$.

Hence,

$$
\begin{aligned}
&\|\left((L^\top L + \lambda I)(\Sigma_\star + \lambda I)^{-1}\right)^{\frac{1}{2}} - \left((L_\star^\top L_\star + \lambda I)(\Sigma_\star + \lambda I)^{-1}\right)^{\frac{1}{2}}\|_{op} \\
&\leq \|(L^\top L + \lambda I)(\Sigma_\star + \lambda I)^{-1} - (L_\star^\top L_\star + \lambda I)(\Sigma_\star + \lambda I)^{-1}\|_{op}^{\frac{1}{2}} = \|\left(L^\top L - L_\star^\top L_\star\right)(\Sigma_\star + \lambda I)^{-1}\|_{op}^{\frac{1}{2}} \\
&= \|\left(L^\top L - L^\top L_\star + L^\top L_\star - L_\star^\top L_\star\right)(\Sigma_\star + \lambda I)^{-1}\|_{op}^{\frac{1}{2}} = \|\left(L^\top (L - L_\star) + (L - L_\star)^\top L_\star\right)(\Sigma_\star + \lambda I)^{-1}\|_{op}^{\frac{1}{2}} \\
&= \|\left((L - L_\star + L_\star)^\top (L - L_\star) + (L - L_\star)^\top L_\star\right)(\Sigma_\star + \lambda I)^{-1}\|_{op}^{\frac{1}{2}} \\
&= \|\left((L - L_\star)^\top (L - L_\star) + L_\star^\top (L - L_\star) + (L - L_\star)^\top L_\star\right)(\Sigma_\star + \lambda I)^{-1}\|_{op}^{\frac{1}{2}} \\
&\leq \|\left((L - L_\star)^\top (L - L_\star) + L_\star^\top (L - L_\star) + (L - L_\star)^\top L_\star\right)\|_{op}^{\frac{1}{2}}\|(\Sigma_\star + \lambda I)^{-1}\|_{op}^{\frac{1}{2}} \\
&\leq \frac{1}{\lambda^{\frac{1}{2}}} \left(\|L - L_\star\|_{op} + \sqrt{2}\|L_\star\|_{op}^{\frac{1}{2}}\|(L - L_\star)\|_{op}^{\frac{1}{2}}\right) \\
&\leq \frac{1}{\lambda^{\frac{1}{2}}} \left(\lambda^{\frac{1}{2}}\epsilon + \sqrt{2}\|X_\star\|_{op}^{\frac{1}{4}}\lambda^{\frac{1}{4}}\sqrt{\epsilon}\right) \\
&= \epsilon + \sqrt{2}\sqrt{\epsilon}\|X_\star\|_{op}^{\frac{1}{4}}\lambda^{-\frac{1}{4}} = \sqrt{\epsilon}(\sqrt{\epsilon} + \sqrt{2}\bar\lambda^{\frac{1}{4}}).
\end{aligned}
$$

From the last inequality, we get that,

$$\|(\boldsymbol{\Sigma}_\star + \lambda \boldsymbol{I})^{-\frac{1}{2}} \Delta_\lambda \boldsymbol{L}\|_{op} \leq \sqrt{\epsilon}(\sqrt{\epsilon} + \sqrt{2}\bar{\lambda}^{\frac{1}{4}}). \tag{38}$$

The proof for the term t$\| (\boldsymbol{\Sigma}_\star + \lambda \boldsymbol{I})^{-\frac{1}{2}} \Delta_\lambda \boldsymbol{R}\|_{op}$ can be derived following similar steps. $\qquad \square$

**Lemma A.2.5.** *Let* $\boldsymbol{L} \in \mathbb{R}^{m \times d}$, $\boldsymbol{R} \in \mathbb{R}^{n \times d}$ *and denote* $\Delta_\lambda \boldsymbol{R} = (\boldsymbol{R}^\top \boldsymbol{R} + \lambda \boldsymbol{I})^{1/2} - (\boldsymbol{R}_\star^\top \boldsymbol{R}_\star + \lambda \boldsymbol{I})^{1/2}$, $\Delta_\lambda \boldsymbol{L} = (\boldsymbol{L}^\top \boldsymbol{L} + \lambda \boldsymbol{I})^{1/2} - (\boldsymbol{L}_\star^\top \boldsymbol{L}_\star + \lambda \boldsymbol{I})^{1/2}$. *Assume that* $\max\{\|\|(\boldsymbol{L} - \boldsymbol{L}_\star)(\boldsymbol{\Sigma}_\star + \lambda \boldsymbol{I})^{-\frac{1}{2}}\|_{op}, \|(\boldsymbol{R} - \boldsymbol{R}_\star)(\boldsymbol{\Sigma}_\star + \lambda \boldsymbol{I})^{-\frac{1}{2}}\|_{op}\} \leq \epsilon$.
*Then it holds,*

$$\| \left(\boldsymbol{R}^\top \boldsymbol{R} + \lambda \boldsymbol{I}\right)^{-\frac{1}{2}} \left(\boldsymbol{\Sigma}_\star + \lambda \boldsymbol{I}\right)^{\frac{1}{2}} \|_{op} \leq \frac{1}{1 - \sqrt{\epsilon}(\sqrt{\epsilon} + \sqrt{2}\bar{\lambda}^{\frac{1}{4}})},$$
$$\| \left(\boldsymbol{L}^\top \boldsymbol{L} + \lambda \boldsymbol{I}\right)^{-\frac{1}{2}} \left(\boldsymbol{\Sigma}_\star + \lambda \boldsymbol{I}\right)^{\frac{1}{2}} \|_{op} \leq \frac{1}{1 - \sqrt{\epsilon}(\sqrt{\epsilon} + \sqrt{2}\bar{\lambda}^{\frac{1}{4}})}. \tag{39}$$

*Proof:* We have $\left(\boldsymbol{R}^\top \boldsymbol{R} + \lambda \boldsymbol{I}\right)^{-\frac{1}{2}} \left(\boldsymbol{\Sigma}_\star + \lambda \boldsymbol{I}\right)^{\frac{1}{2}} = \left((\boldsymbol{\Sigma}_\star + \lambda \boldsymbol{I})^{-\frac{1}{2}} \left(\boldsymbol{R}^\top \boldsymbol{R} + \lambda \boldsymbol{I}\right)^{\frac{1}{2}}\right)^{-1}$. Hence,

$$\| \left(\boldsymbol{R}^\top \boldsymbol{R} + \lambda \boldsymbol{I}\right)^{-\frac{1}{2}} \left(\boldsymbol{\Sigma}_\star + \lambda \boldsymbol{I}\right)^{\frac{1}{2}} \|_{op} = \frac{1}{\sigma_d \left((\boldsymbol{\Sigma}_\star + \lambda \boldsymbol{I})^{-\frac{1}{2}} \left(\boldsymbol{R}^\top \boldsymbol{R} + \lambda \boldsymbol{I}\right)^{\frac{1}{2}}\right)}. \tag{40}$$

By Weyl's inequality, we get

$$\begin{aligned}
&\sigma_d \left((\boldsymbol{\Sigma}_\star + \lambda \boldsymbol{I})^{-\frac{1}{2}} \left(\boldsymbol{R}^\top \boldsymbol{R} + \lambda \boldsymbol{I}\right)^{\frac{1}{2}}\right) \\
&\geq \sigma_d \left((\boldsymbol{\Sigma}_\star + \lambda \boldsymbol{I})^{-\frac{1}{2}} \left(\boldsymbol{R}_\star^\top \boldsymbol{R}_\star + \lambda \boldsymbol{I}\right)^{\frac{1}{2}}\right) - \|(\boldsymbol{\Sigma}_\star + \lambda \boldsymbol{I})^{-\frac{1}{2}} \Delta_\lambda \boldsymbol{R}\|_{op} \\
&= 1 - \|(\boldsymbol{\Sigma}_\star + \lambda \boldsymbol{I})^{-\frac{1}{2}} \Delta_\lambda \boldsymbol{R}\|_{op}.
\end{aligned} \tag{41}$$

From Lemma A.2.4, we have

$$\|(\boldsymbol{\Sigma}_\star + \lambda \boldsymbol{I})^{-\frac{1}{2}} \Delta_\lambda \boldsymbol{R}\|_{op} \leq \sqrt{\epsilon}(\sqrt{\epsilon} + \sqrt{2}\bar{\lambda}^{\frac{1}{4}}). \tag{42}$$

Hence, we conclude the proof. The proof for $\| (\boldsymbol{\Sigma}_\star + \lambda \boldsymbol{I})^{-\frac{1}{2}} \Delta_\lambda \boldsymbol{R}\|_{op}$ can be similarly derived. $\qquad \square$

## A.3. Proof of convergence of OPSA

**Theorem A.3.1** (Convergence of Overparameterized Preconditioned Subgradient Algorithm). *Let* $\mathcal{L}(\boldsymbol{X}) : \mathbb{R}^{m \times n} \mapsto \mathbb{R}$ *be convex w.r.t* $\boldsymbol{X}$, *and assume that it satisfies the rank-d restricted L-Lipschitz continuity assumption and the rank-d restricted* $\mu$-sharpness condition, defined above. Assume also that

$$\mathrm{dist}(\boldsymbol{F}_0, \boldsymbol{F}_\star) \leq \lambda \epsilon \tag{43}$$

*and let* $\lambda = \frac{\|\boldsymbol{X}_\star\|_{op}}{\bar{\lambda}}$, $\sigma_r(\boldsymbol{X}_\star) = 1$, *and* $\chi = \frac{L}{\mu}$. *Then for the Overparameterized Preconditioned Subgradient algorithm given in Algorithm 1, with the Polyak's stepsize defined in* (24), *we have,*

$$\mathrm{dist}(\boldsymbol{F}_t, \boldsymbol{F}_\star) \leq \rho(\chi, \epsilon, \lambda)^{\frac{t}{2}} \left(\frac{\|\boldsymbol{X}\|_{op}}{\bar{\lambda}} \epsilon\right)^{\frac{1}{2}}, \tag{44}$$

*where* $\rho(\chi, \epsilon, \lambda)$ *is given by*

$$\begin{aligned}
\rho(\chi, \epsilon, \lambda) = 1 - \frac{1}{2\chi^2} &\sqrt{\frac{\sqrt{2}-1}{1+2\lambda}} \left(\sqrt{\frac{\sqrt{2}-1}{1+2\lambda}} \left(2 - \frac{1}{(1 - \sqrt{\epsilon}(\sqrt{\epsilon} + \sqrt{2}\bar{\lambda}^{\frac{1}{4}}))^2}\right) \right. \\
&\left. - 2\chi\sqrt{2} \left(\frac{3}{2}\epsilon + 2\frac{\sqrt{\epsilon}(\sqrt{\epsilon} + \sqrt{2}\bar{\lambda}^{\frac{1}{4}})}{1 - \sqrt{\epsilon}(\sqrt{\epsilon} + \sqrt{2}\bar{\lambda}^{\frac{1}{4}})}\right)\right).
\end{aligned} \tag{45}$$

Proof: We denote $\boldsymbol{L} = \boldsymbol{L}_t \boldsymbol{Q}_t, \boldsymbol{R} = \boldsymbol{R}_t \boldsymbol{Q}_t^{-\top}, \Delta_{\boldsymbol{L}} = \boldsymbol{L} - \boldsymbol{L}_\star, \Delta_{\boldsymbol{R}} = \boldsymbol{R} - \boldsymbol{R}_\star$ and $\boldsymbol{S} = \boldsymbol{S}_t$.

We focus on the contraction of dist $(\boldsymbol{F}_{t+1}, \boldsymbol{F}_\star)$.

$$\text{dist}^2 (\boldsymbol{F}_{t+1}, \boldsymbol{F}_\star) \leq \|(\boldsymbol{L}_{t+1} \boldsymbol{Q}_t - \boldsymbol{L}_\star)(\boldsymbol{\Sigma}_\star + \lambda \boldsymbol{I})^{\frac{1}{2}}\|_F^2 + \|(\boldsymbol{R}_{t+1} \boldsymbol{Q}_t^{-\top} - \boldsymbol{R}_\star)(\boldsymbol{\Sigma}_\star + \lambda \boldsymbol{I})^{\frac{1}{2}}\|_F^2. \tag{46}$$

Let $\boldsymbol{L} = \boldsymbol{L}_t \boldsymbol{Q}_t, \boldsymbol{R} = \boldsymbol{R}_t \boldsymbol{Q}_t^{-\top}$ and $\boldsymbol{S} = \boldsymbol{S}_t$ and first bound the term $\|(\boldsymbol{L}_{t+1} \boldsymbol{Q}_t - \boldsymbol{L}_\star)(\boldsymbol{\Sigma}_\star + \lambda \boldsymbol{I})^{\frac{1}{2}}\|_F^2$ as follows,

$$\|(\boldsymbol{L}_{t+1} \boldsymbol{Q}_t - \boldsymbol{L}_\star)(\boldsymbol{\Sigma}_\star + \lambda \boldsymbol{I})^{\frac{1}{2}}\|_F^2 = \| \left( \boldsymbol{L} - \eta \boldsymbol{S} \boldsymbol{R} \left( \boldsymbol{R}^\top \boldsymbol{R} + \lambda \boldsymbol{I} \right)^{-1} - \boldsymbol{L}_\star \right) (\boldsymbol{\Sigma}_\star + \lambda \boldsymbol{I})^{\frac{1}{2}} \|_F^2$$

$$= \|\Delta_{\boldsymbol{L}} (\boldsymbol{\Sigma}_\star + \lambda \boldsymbol{I})^{\frac{1}{2}}\|_F^2 - 2\eta \langle \boldsymbol{S}, \Delta_{\boldsymbol{L}} (\boldsymbol{\Sigma}_\star + \lambda \boldsymbol{I}) \left( \boldsymbol{R}^\top \boldsymbol{R} + \lambda \boldsymbol{I} \right)^{-1} \boldsymbol{R}^\top \rangle$$

$$+ \eta^2 \|\boldsymbol{S} \boldsymbol{R} \left( \boldsymbol{R}^\top \boldsymbol{R} + \lambda \boldsymbol{I} \right)^{-1} (\boldsymbol{\Sigma}_\star + \lambda \boldsymbol{I})^{\frac{1}{2}}\|_F^2$$

$$= \|\Delta_{\boldsymbol{L}} (\boldsymbol{\Sigma}_\star + \lambda \boldsymbol{I})^{\frac{1}{2}}\|_F^2 - 2\eta \underbrace{\langle \boldsymbol{S}, \Delta_{\boldsymbol{L}} \boldsymbol{R}_\star^\top + \frac{1}{2} \Delta_{\boldsymbol{L}} \Delta_{\boldsymbol{R}}^\top \rangle}_{\mathcal{O}_1}$$

$$- 2\eta \underbrace{\langle \boldsymbol{S}, \Delta_{\boldsymbol{L}} (\boldsymbol{\Sigma}_\star + \lambda \boldsymbol{I}) \left( \boldsymbol{R}^\top \boldsymbol{R} + \lambda \boldsymbol{I} \right)^{-1} \boldsymbol{R}^\top - \Delta_{\boldsymbol{L}} \boldsymbol{R}_\star^\top - \frac{1}{2} \Delta_{\boldsymbol{L}} \Delta_{\boldsymbol{R}}^\top \rangle}_{\mathcal{O}_2}$$

$$+ \eta^2 \underbrace{\|\boldsymbol{S} \boldsymbol{R} \left( \boldsymbol{R}^\top \boldsymbol{R} + \lambda \boldsymbol{I} \right)^{-1} (\boldsymbol{\Sigma}_\star + \lambda \boldsymbol{I})^{\frac{1}{2}}\|_F^2}_{\mathcal{O}_3}.$$

We focus on bounding the term $|\mathcal{O}_2|$,

$$|\langle \boldsymbol{S}, \Delta_{\boldsymbol{L}} (\boldsymbol{\Sigma}_\star + \lambda \boldsymbol{I}) \left( \boldsymbol{R}^\top \boldsymbol{R} + \lambda \boldsymbol{I} \right)^{-1} \boldsymbol{R}^\top - \Delta_{\boldsymbol{L}} \boldsymbol{R}_\star^\top - \frac{1}{2} \Delta_{\boldsymbol{L}} \Delta_{\boldsymbol{R}}^\top \rangle|$$

$$\leq \|\boldsymbol{S}\|_F \|\Delta_{\boldsymbol{L}} (\boldsymbol{\Sigma}_\star + \lambda \boldsymbol{I})^{\frac{1}{2}}\|_F \times$$

$$(\|\boldsymbol{R} \left( \boldsymbol{R}^\top \boldsymbol{R} + \lambda \boldsymbol{I} \right)^{-1} (\boldsymbol{\Sigma}_\star + \lambda \boldsymbol{I})^{\frac{1}{2}} - \boldsymbol{R}_\star (\boldsymbol{\Sigma}_\star + \lambda \boldsymbol{I})^{-\frac{1}{2}}\|_{op} + \frac{1}{2} \|\Delta_{\boldsymbol{R}} (\boldsymbol{\Sigma}_\star + \lambda \boldsymbol{I})^{-\frac{1}{2}}\|_{op}), \tag{47}$$

where we have used the inequalities $\|\boldsymbol{A}\boldsymbol{B}\|_F \leq \|\boldsymbol{A}\|_{op}\|\boldsymbol{B}\|_F$ and $\|\boldsymbol{A} - \boldsymbol{B}\|_{op} \leq \|\boldsymbol{A}\|_{op} + \|\boldsymbol{B}\|_{op}$.

For the first term of (47) we have

$$\|\boldsymbol{R} \left( \boldsymbol{R}^\top \boldsymbol{R} + \lambda \boldsymbol{I} \right)^{-1} (\boldsymbol{\Sigma}_\star + \lambda \boldsymbol{I})^{\frac{1}{2}} - \boldsymbol{R}_\star (\boldsymbol{\Sigma}_\star + \lambda \boldsymbol{I})^{-\frac{1}{2}}\|_{op}$$

$$= \|\boldsymbol{R} \left( \boldsymbol{R}^\top \boldsymbol{R} + \lambda \boldsymbol{I} \right)^{-\frac{1}{2}} \left( \boldsymbol{R}^\top \boldsymbol{R} + \lambda \boldsymbol{I} \right)^{-\frac{1}{2}} (\boldsymbol{\Sigma}_\star + \lambda \boldsymbol{I})^{\frac{1}{2}} - \boldsymbol{R}_\star (\boldsymbol{\Sigma}_\star + \lambda \boldsymbol{I})^{-\frac{1}{2}}\|_{op}$$

$$= \|\boldsymbol{R} \left( \boldsymbol{R}^\top \boldsymbol{R} + \lambda \boldsymbol{I} \right)^{-\frac{1}{2}} \left( \left( \left( \boldsymbol{R}^\top \boldsymbol{R} + \lambda \boldsymbol{I} \right)^{-\frac{1}{2}} - \left( \boldsymbol{R}_\star^\top \boldsymbol{R}_\star + \lambda \boldsymbol{I} \right)^{-\frac{1}{2}} \right) (\boldsymbol{\Sigma}_\star + \lambda \boldsymbol{I})^{\frac{1}{2}} + \left( \boldsymbol{R}_\star^\top \boldsymbol{R}_\star + \lambda \boldsymbol{I} \right)^{-\frac{1}{2}} (\boldsymbol{\Sigma}_\star + \lambda \boldsymbol{I})^{\frac{1}{2}} \right)$$

$$- \boldsymbol{R}_\star (\boldsymbol{\Sigma}_\star + \lambda \boldsymbol{I})^{-\frac{1}{2}}\|_{op}$$

$$= \|\boldsymbol{R} \left( \boldsymbol{R}^\top \boldsymbol{R} + \lambda \boldsymbol{I} \right)^{-\frac{1}{2}} \left( \left( \left( \boldsymbol{R}^\top \boldsymbol{R} + \lambda \boldsymbol{I} \right)^{-\frac{1}{2}} - \left( \boldsymbol{R}_\star^\top \boldsymbol{R}_\star + \lambda \boldsymbol{I} \right)^{-\frac{1}{2}} \right) (\boldsymbol{\Sigma}_\star + \lambda \boldsymbol{I})^{\frac{1}{2}} + \boldsymbol{I} \right) - \boldsymbol{R}_\star (\boldsymbol{\Sigma}_\star + \lambda \boldsymbol{I})^{-\frac{1}{2}}\|_{op}$$

$$\leq \|\boldsymbol{R} \left( \boldsymbol{R}^\top \boldsymbol{R} + \lambda \boldsymbol{I} \right)^{-\frac{1}{2}}\|_{op} \left\| \left( \left( \boldsymbol{R}^\top \boldsymbol{R} + \lambda \boldsymbol{I} \right)^{-\frac{1}{2}} - \left( \boldsymbol{R}_\star^\top \boldsymbol{R}_\star + \lambda \boldsymbol{I} \right)^{-\frac{1}{2}} \right) (\boldsymbol{\Sigma}_\star + \lambda \boldsymbol{I})^{\frac{1}{2}} \right\|_{op}$$

$$+ \|\boldsymbol{R} \left( \boldsymbol{R}^\top \boldsymbol{R} + \lambda \boldsymbol{I} \right)^{-\frac{1}{2}} - \boldsymbol{R}_\star \left( \boldsymbol{R}_\star^\top \boldsymbol{R}_\star + \lambda \boldsymbol{I} \right)^{-\frac{1}{2}}\|_{op}.$$

We now focus on bounding the term $\|\left( \left( \boldsymbol{R}^\top \boldsymbol{R} + \lambda \boldsymbol{I} \right)^{-\frac{1}{2}} - \left( \boldsymbol{R}_\star^\top \boldsymbol{R}_\star + \lambda \boldsymbol{I} \right)^{-\frac{1}{2}} \right) (\boldsymbol{\Sigma}_\star + \lambda \boldsymbol{I})^{\frac{1}{2}}\|_{op}$. Let $\boldsymbol{A} = \boldsymbol{R}^\top \boldsymbol{R} + \lambda \boldsymbol{I}$

and $\boldsymbol{B} = \boldsymbol{R}_\star^\top \boldsymbol{R}_\star + \lambda \boldsymbol{I}$. We have

$$
\begin{aligned}
&\| (\boldsymbol{A}^{-\frac{1}{2}} - \boldsymbol{B}^{-\frac{1}{2}}) (\boldsymbol{\Sigma}_\star + \lambda \boldsymbol{I})^{\frac{1}{2}} \|_{op} = \| (\boldsymbol{A}^{-\frac{1}{2}} - \boldsymbol{B}^{-\frac{1}{2}}) (\boldsymbol{A}^{\frac{1}{2}} \boldsymbol{B}^{\frac{1}{2}}) (\boldsymbol{A}^{\frac{1}{2}} \boldsymbol{B}^{\frac{1}{2}})^{-1} (\boldsymbol{\Sigma}_\star + \lambda \boldsymbol{I})^{\frac{1}{2}} \|_{op} \\
&= \| (\boldsymbol{B}^{\frac{1}{2}} - \boldsymbol{B}^{-\frac{1}{2}} \boldsymbol{A}^{\frac{1}{2}} \boldsymbol{B}^{\frac{1}{2}}) (\boldsymbol{A}^{\frac{1}{2}} \boldsymbol{B}^{\frac{1}{2}})^{-1} (\boldsymbol{\Sigma}_\star + \lambda \boldsymbol{I})^{\frac{1}{2}} \|_{op} \\
&= \| (\boldsymbol{B}^{\frac{1}{2}} - \boldsymbol{B}^{-\frac{1}{2}} \boldsymbol{A}^{\frac{1}{2}} \boldsymbol{B}^{\frac{1}{2}}) \boldsymbol{B}^{-\frac{1}{2}} (\boldsymbol{A}^{-\frac{1}{2}}) (\boldsymbol{\Sigma}_\star + \lambda \boldsymbol{I})^{\frac{1}{2}} \|_{op} \\
&= \| (\boldsymbol{I} - \boldsymbol{B}^{-\frac{1}{2}} \boldsymbol{A}^{\frac{1}{2}}) (\boldsymbol{A}^{-\frac{1}{2}}) (\boldsymbol{\Sigma}_\star + \lambda \boldsymbol{I})^{\frac{1}{2}} \|_{op} \\
&\leq \| \boldsymbol{B}^{-\frac{1}{2}} \left( \boldsymbol{B}^{\frac{1}{2}} - \boldsymbol{A}^{\frac{1}{2}} \right) \|_{op} \| \boldsymbol{A}^{-\frac{1}{2}} (\boldsymbol{\Sigma}_\star + \lambda \boldsymbol{I})^{\frac{1}{2}} \|_{op} \\
&= \frac{\sqrt{\epsilon} (\sqrt{\epsilon} + \sqrt{2} \bar{\lambda}^{\frac{1}{4}})}{1 - \sqrt{\epsilon} (\sqrt{\epsilon} + \sqrt{2} \bar{\lambda}^{\frac{1}{4}})},
\end{aligned}
\tag{48}
$$

where the last inequality follows by using Lemmata A.2.4 and A.2.5.

Next, we bound the term $\| \boldsymbol{R} \left( \boldsymbol{R}^\top \boldsymbol{R} + \lambda \boldsymbol{I} \right)^{-\frac{1}{2}} - \boldsymbol{R}_\star \left( \boldsymbol{R}_\star^\top \boldsymbol{R}_\star + \lambda \boldsymbol{I} \right)^{-\frac{1}{2}} \|_{op}$. We have

$$
\begin{aligned}
&\| \boldsymbol{R} \left( \boldsymbol{R}^\top \boldsymbol{R} + \lambda \boldsymbol{I} \right)^{-\frac{1}{2}} - \boldsymbol{R}_\star \left( \boldsymbol{R}_\star^\top \boldsymbol{R}_\star + \lambda \boldsymbol{I} \right)^{-\frac{1}{2}} \|_{op} \\
&= \| (\boldsymbol{R} - \boldsymbol{R}_\star) \left( \boldsymbol{R}^\top \boldsymbol{R} + \lambda \boldsymbol{I} \right)^{-\frac{1}{2}} + \boldsymbol{R}_\star \left( \left( \boldsymbol{R}^\top \boldsymbol{R} + \lambda \boldsymbol{I} \right)^{-\frac{1}{2}} - \left( \boldsymbol{R}_\star^\top \boldsymbol{R}_\star + \lambda \boldsymbol{I} \right)^{-\frac{1}{2}} \right) \|_{op} \\
&\leq \| (\boldsymbol{R} - \boldsymbol{R}_\star) \left( \boldsymbol{R}^\top \boldsymbol{R} + \lambda \boldsymbol{I} \right)^{-\frac{1}{2}} \|_{op} + \| \boldsymbol{R}_\star \left( \left( \boldsymbol{R}^\top \boldsymbol{R} + \lambda \boldsymbol{I} \right)^{-\frac{1}{2}} - \left( \boldsymbol{R}_\star^\top \boldsymbol{R}_\star + \lambda \boldsymbol{I} \right)^{-\frac{1}{2}} \right) \|_{op} \\
&\leq \| \boldsymbol{R} - \boldsymbol{R}_\star \|_{op} \| \left( \boldsymbol{R}^\top \boldsymbol{R} + \lambda \boldsymbol{I} \right)^{-\frac{1}{2}} \|_{op} + \| \boldsymbol{R}_\star \left( \left( \boldsymbol{R}^\top \boldsymbol{R} + \lambda \boldsymbol{I} \right)^{-\frac{1}{2}} - \left( \boldsymbol{R}_\star^\top \boldsymbol{R}_\star + \lambda \boldsymbol{I} \right)^{-\frac{1}{2}} \right) \|_{op} \\
&\leq \epsilon + \frac{\sqrt{\epsilon} (\sqrt{\epsilon} + \sqrt{2} \bar{\lambda}^{\frac{1}{4}})}{1 - \sqrt{\epsilon} (\sqrt{\epsilon} + \sqrt{2} \bar{\lambda}^{\frac{1}{4}})},
\end{aligned}
\tag{49}
$$

where we have used the inequality $\| \boldsymbol{R} - \boldsymbol{R}_\star \|_{op} \leq \epsilon \sqrt{\lambda}$. Note that this result holds, since we assumed $\mathrm{dist}(\boldsymbol{F}, \boldsymbol{F}_\star) \leq \lambda \epsilon$, hence we can show the following,

$$
\| (\boldsymbol{L} - \boldsymbol{L}_\star) (\boldsymbol{\Sigma}_\star + \lambda \boldsymbol{I})^{\frac{1}{2}}) (\boldsymbol{\Sigma}_\star + \lambda \boldsymbol{I})^{-\frac{1}{2}}) \|_{op}^2 \sigma_r^2 ((\boldsymbol{\Sigma}_\star + \lambda \boldsymbol{I})^{\frac{1}{2}}))
\tag{50}
$$

$$
+ \| (\boldsymbol{R} - \boldsymbol{R}_\star) (\boldsymbol{\Sigma}_\star + \lambda \boldsymbol{I})^{\frac{1}{2}}) (\boldsymbol{\Sigma}_\star + \lambda \boldsymbol{I})^{\frac{1}{2}}) \|_{op}^2 \sigma_r^2 ((\boldsymbol{\Sigma}_\star + \lambda \boldsymbol{I})^{\frac{1}{2}})) < \lambda^2 \epsilon^2
$$

$$
\rightarrow \| (\boldsymbol{L} - \boldsymbol{L}_\star) (\boldsymbol{\Sigma}_\star + \lambda \boldsymbol{I})^{\frac{1}{2}}) (\boldsymbol{\Sigma}_\star + \lambda \boldsymbol{I})^{-\frac{1}{2}}) \|_{op}^2 \lambda + \| (\boldsymbol{R} - \boldsymbol{R}_\star) (\boldsymbol{\Sigma}_\star + \lambda \boldsymbol{I})^{\frac{1}{2}}) (\boldsymbol{\Sigma}_\star + \lambda \boldsymbol{I})^{\frac{1}{2}} 2) \|_{op}^2 \sigma_r ((\boldsymbol{\Sigma}_\star + \lambda \boldsymbol{I})^{\frac{1}{2}})) < \lambda^2 \epsilon^2
$$

$$
\max \left( \| \boldsymbol{L} - \boldsymbol{L}_\star \|_{op}, \| \boldsymbol{R} - \boldsymbol{R}_\star \|_{op} \right) < \sqrt{\lambda} \epsilon.
\tag{51}
$$

Combining the above bounds we get

$$
\begin{aligned}
&\| \boldsymbol{R} \left( \boldsymbol{R}^\top \boldsymbol{R} + \lambda \boldsymbol{I} \right)^{-1} (\boldsymbol{\Sigma}_\star + \lambda \boldsymbol{I})^{\frac{1}{2}} - \boldsymbol{R}_\star (\boldsymbol{\Sigma}_\star + \lambda \boldsymbol{I})^{-\frac{1}{2}} \|_{op} \\
&\leq \epsilon + 2 \frac{\sqrt{\epsilon} (\sqrt{\epsilon} + \sqrt{2} \bar{\lambda}^{\frac{1}{4}})}{1 - \sqrt{\epsilon} (\sqrt{\epsilon} + \sqrt{2} \bar{\lambda}^{\frac{1}{4}})}.
\end{aligned}
\tag{52}
$$

We now focus on bounding $\mathcal{O}_3$,

$$
\begin{aligned}
&\| \boldsymbol{S} \boldsymbol{R} \left( \boldsymbol{R}^\top \boldsymbol{R} + \lambda \boldsymbol{I} \right)^{-1} (\boldsymbol{\Sigma}_\star + \lambda \boldsymbol{I})^{\frac{1}{2}} \|_F^2 \\
&\leq \underbrace{\| \boldsymbol{S} \boldsymbol{R} \left( \boldsymbol{R}^\top \boldsymbol{R} + \lambda \boldsymbol{I} \right)^{-\frac{1}{2}} \|_F^2}_{\mathcal{O}_{3a}} \times \underbrace{\| \left( \boldsymbol{R}^\top \boldsymbol{R} + \lambda \boldsymbol{I} \right)^{-\frac{1}{2}} (\boldsymbol{\Sigma}_\star + \lambda \boldsymbol{I})^{\frac{1}{2}} \|_{op}^2}_{\mathcal{O}_{3b}} \\
&\leq \| \boldsymbol{S} \boldsymbol{R} \left( \boldsymbol{R}^\top \boldsymbol{R} + \lambda \boldsymbol{I} \right)^{-\frac{1}{2}} \|_F^2 \frac{1}{(1 - \sqrt{\epsilon} (\sqrt{\epsilon} + \sqrt{2} \bar{\lambda}^{\frac{1}{4}}))^2}.
\end{aligned}
\tag{53}
$$

Using the derived inequalities above we have

$$\|(\boldsymbol{L}_{t+1}\boldsymbol{Q}_t - \boldsymbol{L}_\star)(\boldsymbol{\Sigma}_\star + \lambda\boldsymbol{I})^{\frac{1}{2}}\|_F^2 \le \|\Delta_{\boldsymbol{L}}(\boldsymbol{\Sigma}_\star + \lambda\boldsymbol{I})^{\frac{1}{2}}\|_F^2 - 2\eta\langle\boldsymbol{S}, \Delta_{\boldsymbol{L}}\boldsymbol{R}_\star^\top + \frac{1}{2}\Delta_{\boldsymbol{L}}\Delta_{\boldsymbol{R}}^\top\rangle$$
$$+ 2\eta L\|\Delta_{\boldsymbol{L}}(\boldsymbol{\Sigma}_\star + \lambda\boldsymbol{I})^{\frac{1}{2}}\|_F\left(\frac{3}{2}\epsilon + 2\frac{\sqrt{\epsilon}(\sqrt{\epsilon} + \sqrt{2}\bar{\lambda}^{\frac{1}{4}})}{1 - \left(\sqrt{\epsilon}(\sqrt{\epsilon} + \sqrt{2}\bar{\lambda}^{\frac{1}{4}})\right)}\right)$$
$$+ \eta^2\frac{1}{(1 - \sqrt{\epsilon}(\sqrt{\epsilon} + \sqrt{2}\bar{\lambda}^{\frac{1}{4}}))^2}\|\boldsymbol{S}\boldsymbol{R}(\boldsymbol{R}^\top\boldsymbol{R} + \lambda\boldsymbol{I})\|_F^2.$$

Following the same steps for bounding the term $\|(\boldsymbol{R}_{t+1}\boldsymbol{Q}_t^{-\top} - \boldsymbol{R}_\star)(\boldsymbol{\Sigma}_\star + \lambda\boldsymbol{I})^{\frac{1}{2}}\|_F^2$, we have

$$\text{dist}^2(\boldsymbol{F}_{t+1}, \boldsymbol{F}_\star)$$
$$\le \|\Delta_{\boldsymbol{L}}(\boldsymbol{\Sigma}_\star + \lambda\boldsymbol{I})^{\frac{1}{2}}\|_F^2 + \|\Delta_{\boldsymbol{R}}(\boldsymbol{\Sigma}_\star + \lambda\boldsymbol{I})^{\frac{1}{2}}\|_F^2 - 2\eta\langle\boldsymbol{S}, \boldsymbol{X} - \boldsymbol{X}_\star\rangle$$
$$+ 2\eta L\left(\frac{3}{2}\epsilon + 2\frac{\sqrt{\epsilon}(\sqrt{\epsilon} + \sqrt{2}\bar{\lambda}^{\frac{1}{4}})}{1 - \sqrt{\epsilon}(\sqrt{\epsilon} + \sqrt{2}\bar{\lambda}^{\frac{1}{4}})}\right)\left(\|\Delta_{\boldsymbol{L}}(\boldsymbol{\Sigma}_\star + \lambda\boldsymbol{I})^{\frac{1}{2}}\|_F + \|\Delta_{\boldsymbol{R}}(\boldsymbol{\Sigma}_\star + \lambda\boldsymbol{I})^{\frac{1}{2}}\|_F\right)$$
$$+ \eta^2\frac{1}{(1 - \sqrt{\epsilon}(\sqrt{\epsilon} + \sqrt{2}\bar{\lambda}^{\frac{1}{4}}))^2}\left(\left(\|\boldsymbol{S}\boldsymbol{R}(\boldsymbol{R}^\top\boldsymbol{R} + \lambda\boldsymbol{I})^{-\frac{1}{2}}\|_F^2 + \|\boldsymbol{S}^\top\boldsymbol{L}(\boldsymbol{L}^\top\boldsymbol{L} + \lambda\boldsymbol{I})^{-\frac{1}{2}}\|_F^2\right)\right.$$
$$\le \text{dist}^2(\boldsymbol{F}_t, \boldsymbol{F}_\star) - 2\eta\left(\tilde{\mathcal{L}}(\boldsymbol{L}_t\boldsymbol{R}_t^\top) - \tilde{\mathcal{L}}(\boldsymbol{X}_\star)\right) + 2\eta L\sqrt{2}\left(\frac{3}{2}\epsilon + 2\frac{\sqrt{\epsilon}(\sqrt{\epsilon} + \sqrt{2}\bar{\lambda}^{\frac{1}{4}})}{1 - \sqrt{\epsilon}(\sqrt{\epsilon} + \sqrt{2}\bar{\lambda}^{\frac{1}{4}})}\right)\text{dist}(\boldsymbol{F}_t, \boldsymbol{F}_\star)$$
$$+ \eta^2\frac{1}{(1 - \sqrt{\epsilon}(\sqrt{\epsilon} + \sqrt{2}\bar{\lambda}^{\frac{1}{4}}))^2}\left(\|\boldsymbol{S}\boldsymbol{R}(\boldsymbol{R}^\top\boldsymbol{R} + \lambda\boldsymbol{I})^{-\frac{1}{2}}\|_F^2 + \|\boldsymbol{S}^\top\boldsymbol{L}(\boldsymbol{L}^\top\boldsymbol{L} + \lambda\boldsymbol{I})^{-\frac{1}{2}}\|_F^2\right).$$

**Convergence with Polyak's stepsize.** We use Polyak's stepsize defined as

$$\eta_t = \frac{\tilde{\mathcal{L}}(\boldsymbol{L}_t\boldsymbol{R}_t^\top) - \tilde{\mathcal{L}}(\boldsymbol{X}_\star)}{\|\boldsymbol{S}\boldsymbol{R}(\boldsymbol{R}^\top\boldsymbol{R} + \lambda\boldsymbol{I})^{-\frac{1}{2}}\|_F^2 + \|\boldsymbol{S}^\top\boldsymbol{L}(\boldsymbol{L}^\top\boldsymbol{L} + \lambda\boldsymbol{I})^{-\frac{1}{2}}\|_F^2}. \tag{54}$$

Due to convexity of $\mathcal{L}(\boldsymbol{X})$, Lemma A.2.2, and the assumption $\sigma_r(\boldsymbol{X}_\star) = 1$, we also have,

$$\langle\boldsymbol{S}, \boldsymbol{X} - \boldsymbol{X}_\star\rangle \ge \tilde{\mathcal{L}}(\boldsymbol{X}) - \tilde{\mathcal{L}}(\boldsymbol{X}_\star) \ge \mu\|\boldsymbol{X} - \boldsymbol{X}_\star\| \ge \mu\sqrt{\frac{(\sqrt{2} - 1)}{1 + 2\lambda}}\text{dist}(\boldsymbol{F}, \boldsymbol{F}_\star).$$

We thus get

$$\text{dist}^2(\boldsymbol{F}_{t+1}, \boldsymbol{F}_\star) \le \text{dist}^2(\boldsymbol{F}_t, \boldsymbol{F}_\star) - \eta_t\left(2 - \frac{1}{(1 - \sqrt{\epsilon}(\sqrt{\epsilon} + \sqrt{2}\bar{\lambda}^{\frac{1}{4}}))^2}\right)(\tilde{\mathcal{L}}(\boldsymbol{X}) - \tilde{\mathcal{L}}(\boldsymbol{X}_\star))$$
$$+ 2\sqrt{2}\eta_t L\left(\frac{3}{2}\epsilon + 2\frac{\sqrt{\epsilon}(\sqrt{\epsilon} + \sqrt{2}\bar{\lambda}^{\frac{1}{4}})}{1 - \sqrt{\epsilon}(\sqrt{\epsilon} + \sqrt{2}\bar{\lambda}^{\frac{1}{4}})}\right)\text{dist}(\boldsymbol{F}_t, \boldsymbol{F}_\star)$$
$$\le \text{dist}^2(\boldsymbol{F}_t, \boldsymbol{F}_\star) - \eta_t\mu\left(\left(2 - \frac{1}{(1 - \sqrt{\epsilon}(\sqrt{\epsilon} + \sqrt{2}\bar{\lambda}^{\frac{1}{4}}))^2}\right)\sqrt{\frac{\sqrt{2} - 1}{1 + 2\lambda}}\right.$$
$$\left. - 2\sqrt{2}\frac{L}{\mu}\left(\frac{3}{2}\epsilon + 2\frac{\sqrt{\epsilon}(\sqrt{\epsilon} + \sqrt{2}\bar{\lambda}^{\frac{1}{4}})}{1 - \sqrt{\epsilon}(\sqrt{\epsilon} + \sqrt{2}\bar{\lambda}^{\frac{1}{4}})}\right)\right)\text{dist}(\boldsymbol{F}_t, \boldsymbol{F}_\star).$$

Using (Tong et al., 2021b, Lemma A.2.2 and Lemma 4), we can lower bound the step size as

$$\eta_t \ge \frac{\mu}{2L^2}\sqrt{\frac{\sqrt{2} - 1}{1 + 2\lambda}}\text{dist}(\boldsymbol{F}, \boldsymbol{F}_\star).$$

We define $\chi = \frac{L}{\mu}$ and then we get

$$\text{dist}^2(\boldsymbol{F}_{t+1}, \boldsymbol{F}_\star)$$

$$\leq \left(1 - \frac{1}{2\chi^2}\sqrt{\frac{\sqrt{2}-1}{1+2\lambda}}\left(\sqrt{\frac{\sqrt{2}-1}{1+2\lambda}}\left(2 - \frac{1}{(1-\sqrt{\epsilon}(\sqrt{\epsilon}+\sqrt{2}\bar{\lambda}^{\frac{1}{4}}))^2}\right)\right.\right.$$

$$\left.\left. - 2\chi\sqrt{2}\left(\frac{3}{2}\epsilon + 2\frac{\sqrt{\epsilon}(\sqrt{\epsilon}+\sqrt{2}\bar{\lambda}^{\frac{1}{4}})}{1-\sqrt{\epsilon}(\sqrt{\epsilon}+\sqrt{2}\bar{\lambda}^{\frac{1}{4}})}\right)\right)\right)\text{dist}^2(\boldsymbol{F}_t, \boldsymbol{F}_\star).$$

We define $\rho(\chi, \epsilon, \lambda)$ as the contraction rate given by,

$$\rho(\chi, \epsilon, \lambda) = 1 - \frac{1}{2\chi^2}\sqrt{\frac{\sqrt{2}-1}{1+2\lambda}}\left(\sqrt{\frac{\sqrt{2}-1}{1+2\lambda}}\left(2 - \frac{1}{(1-\sqrt{\epsilon}(\sqrt{\epsilon}+\sqrt{2}\bar{\lambda}^{\frac{1}{4}}))^2}\right)\right.$$

$$\left. - 2\chi\sqrt{2}\left(\frac{3}{2}\epsilon + 2\frac{\sqrt{\epsilon}(\sqrt{\epsilon}+\sqrt{2}\bar{\lambda}^{\frac{1}{4}})}{1-\sqrt{\epsilon}(\sqrt{\epsilon}+\sqrt{2}\bar{\lambda}^{\frac{1}{4}})}\right)\right).$$

This finishes the proof. $\qquad\square$

### A.3.1. Proof of Theorem 5.4

**Theorem 5.4** *Let $\mathcal{L}(\boldsymbol{X}) : \mathbb{R}^{m\times n} \mapsto \mathbb{R}$ be convex w.r.t $\boldsymbol{X}$, and assume that it satisfies the rank-$d$ restricted $L$-Lipschitz continuity assumption and the rank-$d$ restricted $\mu$-sharpness condition, defined above. Assume also that $\lambda = \frac{\|\boldsymbol{X}_\star\|_{op}}{\bar{\lambda}}$, $\bar{\lambda} = c\|\boldsymbol{X}_\star\|_{op}$, $c = 20$, and without loss of generality $\sigma_r(\boldsymbol{X}_\star) = 1$. Set $\epsilon = \frac{10^{-4}}{\chi\|\boldsymbol{X}_\star\|_{op}^{\frac{1}{2}}}$, which implies.*

$$\text{dist}(\boldsymbol{F}_0, \boldsymbol{F}_\star) \leq \lambda\epsilon = \frac{10^{-4}}{\chi}\frac{\|\boldsymbol{X}_\star\|_{op}^{-\frac{1}{2}}}{c}, \tag{55}$$

*where $\chi = \frac{L}{\mu}$. Then for the Overparameterized Preconditioned Subgradient Algorithm (OPSA) given in Algorithm 1, with the Polyak's step-size defined in (24), we have,*

$$\text{dist}(\boldsymbol{F}_t, \boldsymbol{F}_\star) \leq \left(1 - \frac{0.12}{\chi^2}\right)^{\frac{t}{2}}\frac{10^{-4}\|\boldsymbol{X}_\star\|_{op}^{-\frac{1}{2}}}{c\chi},$$

$$\|\boldsymbol{L}_t\boldsymbol{R}_t^\top - \boldsymbol{X}_\star\|_F \leq \left(1 - \frac{0.12}{\chi^2}\right)^{\frac{t}{2}}\frac{1.5 \times 10^{-4}\|\boldsymbol{X}_\star\|_{op}^{-\frac{1}{2}}}{c\chi}.$$

*Proof:* By using the general expression for the rate $\rho(\chi, \epsilon, \lambda)$ provided in Theorem A.3.1 and for $\lambda = \frac{1}{c}$, and $\epsilon = \frac{10^{-4}}{\|\boldsymbol{X}_\star\|_{op}^{\frac{1}{2}}\chi}$. We have,

$$(1 - \rho(\chi, \epsilon, \lambda))\chi^2 \geq 0.12, \tag{56}$$

which leads to the contraction rate showing up in the statement of the Theorem. By applying Lemma A.2.3, we can show the second inequality. $\qquad\square$

## A.4. More Numerical Experiments

### A.4.1. Partially Observed Video Background Subtraction

Video background subtraction with partial observation is one standard benchmark for robust matrix completion. Note that the RIP condition of matrix completion is only defined with the tangent space projection and the incoherence condition. Nevertheless, following the setup as in (Cai et al., 2025), we apply OPSA to the partially observed video background subtraction task on two real video datasets, namely *shoppingmall* and *restaurant*[1]. The visual results are reported in Figures 8 and 9, compared against the fully observed RPCA results. Although the rank is overestimated and the observation is partially accessible, OPSA achieves crisp visual results.

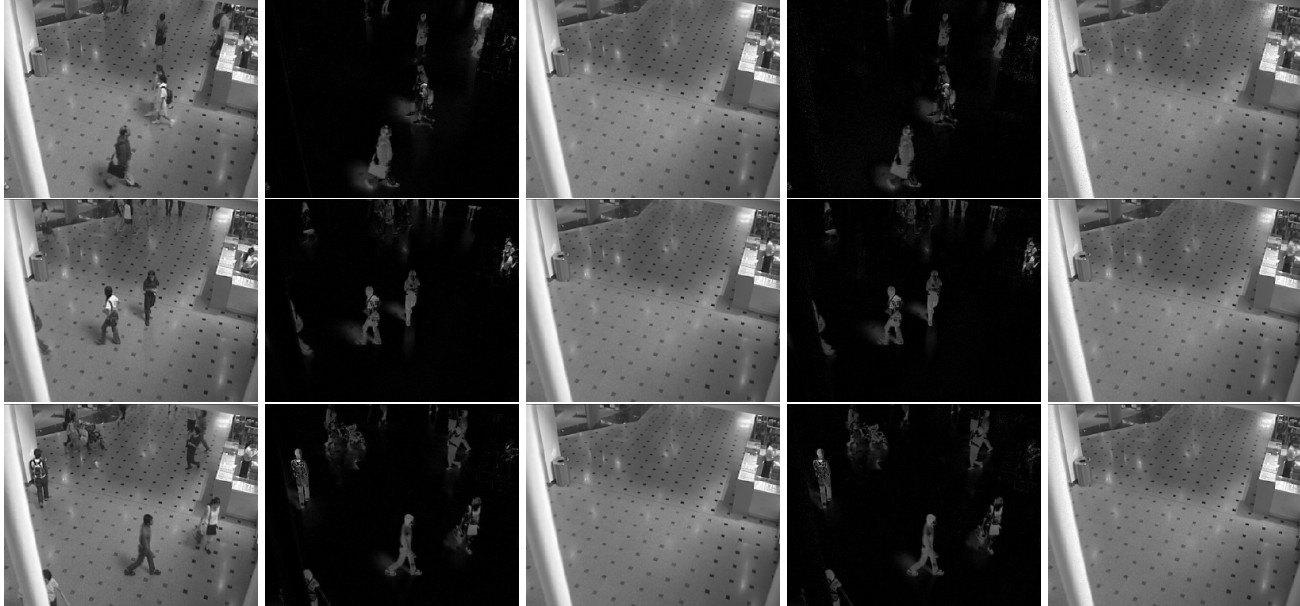

Figure 8. Video background subtraction on *shoppingmall* video. Each row corresponds to a frame in the video. The first column is for the original frames. The next two columns are the "groundtruth" foreground and background provided by AccAltProj (Cai et al., 2019), a non-convex RPCA algorithm, with rank 2 and full observation. The last two columns are the foreground and background outputted by the proposed OPSA with $5\times$ overestimated rank $d = 10$ and $30\%$ observation.

---

[1]The datasets were originally provided by http://perception.i2r.a-star.edu.sg/bk_model/bk_index.html; however, it is no longer available. The datasets are now available at https://hqcai.org/datasets/shoppingmall.mat and https://hqcai.org/datasets/restaurant.mat, respectively.

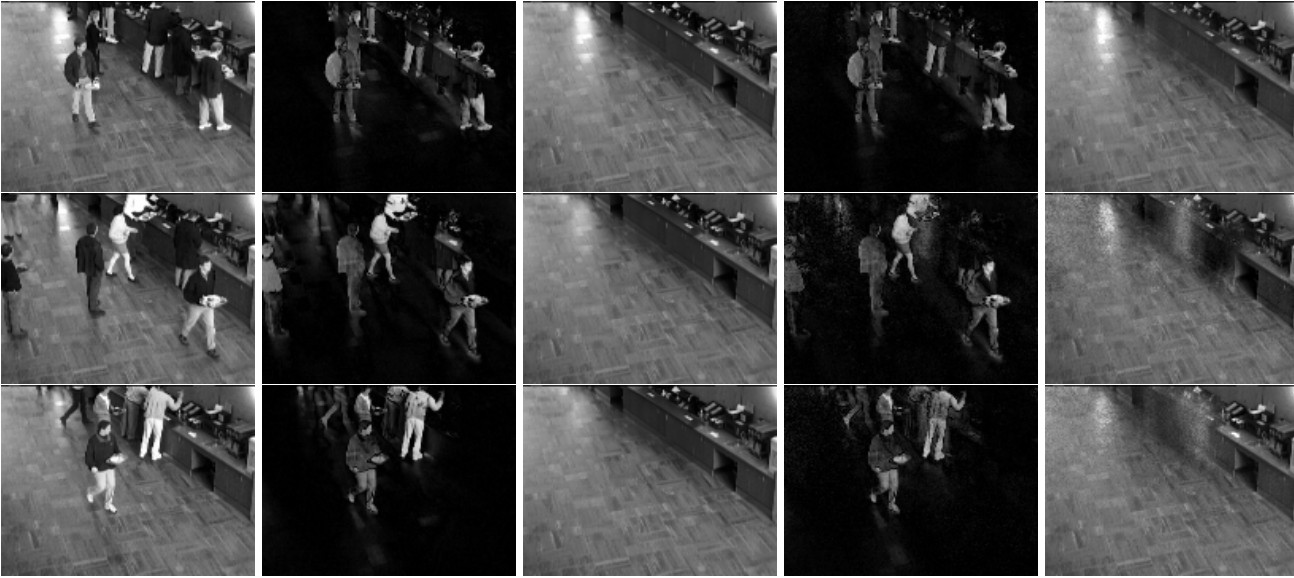

*Figure 9.* Video background subtraction on *restaurant* video. Each row corresponds to a frame in the video. The first column is for the original frames. The next two columns are the "groundtruth" foreground and background provided by AccAltProj (Cai et al., 2019), a non-convex RPCA algorithm, with rank 2 and full observation. The last two columns are the foreground and background outputted by the proposed OPSA with $5\times$ overestimated rank $d = 10$ and $30\%$ observation.

