# OpenReview forum: "Guarantees of a Preconditioned Subgradient Algorithm for Overparameterized Asymmetric Low-rank Matrix Recovery"
_ICML.cc/2025/Conference — ICML 2025 poster_

### Official Review · Reviewer_SuTj · 2025-03-13

**Overall Recommendation:** 2

**Summary:**

This paper establishes theoretical guarantees for the preconditioned subgradient algorithm in the context of overparameterized low-rank matrix recovery (LRMR), with a particular focus on the non-smooth case. It is rigorously proven that the proposed preconditioned subgradient method achieves linear convergence to the true solution. The paper is well-written, with clear and coherent exposition, and presents compelling results that contribute meaningfully to the field.

**Claims And Evidence:**

The authors assert that the convergence rate of the proposed preconditioned subgradient method is independent of the condition number of the target matrix, as stated in Theorem 5.4. However, the experimental results presented in Figure 4 are insufficient to fully validate this claim. Specifically, the relative error in Figure 4 reaches magnitudes as small as 1e-6, which is still too large to conclusively demonstrate the entire convergence process. To strengthen their argument, the authors are encouraged to provide additional experimental evidence showing the relative error at a much finer precision, such as $10^{-14}$, to ensure the robustness of their convergence analysis.

Furthermore, the results in Figure 4 indicate differences in convergence rates across varying condition numbers $\kappa$, where $\kappa$ is set to relatively small values (20, 40, 60, 80, 100). To more rigorously test the independence of the convergence rate from the condition number, the authors should consider conducting experiments with significantly larger values of $\kappa$, such as $10^3$, $10^4$, and beyond. This would provide a more comprehensive evaluation of the method's performance under a wider range of conditions and strengthen the empirical support for their theoretical claims.

**Essential References Not Discussed:**

Not available

**Experimental Designs Or Analyses:**

Regarding the experimental results in Figure 5, similar to Figure 4, the relative error is only shown up to a precision of 1e-6. To strengthen their argument, the authors are encouraged to provide additional experimental evidence demonstrating the relative error at a much finer precision, such as $10^{-14}$. This would offer more robust support for the claimed convergence properties of the algorithm.

Additionally, the authors state in line 435 that with a large value of $\lambda$, OPSA may become trapped in a local minimum. However, this conclusion lacks theoretical solidity. Specifically, for this non-convex optimization problem, if local minima exist, it remains unclear how a gradient/subgradient-based algorithm like OPSA can escape these local minima and converge to the global solution. As observed in Figure 4, the results for $\lambda=10$ show that OPSA has not yet converged, even after the presented number of iterations. To provide a more comprehensive understanding, the authors should extend the iteration count to at least $10^4$ and present the corresponding results. This would help clarify whether OPSA can eventually escape local minima and achieve global convergence under varying conditions.

**Methods And Evaluation Criteria:**

Yes.

**Other Comments Or Suggestions:**

Not available

**Other Strengths And Weaknesses:**

1. The authors claim that the distance metric introduced in Eq. (22) is novel. However, to the best of my knowledge, this appears to be only an incremental improvement over the distance metric proposed by Tong et al., 2021a. While the modification may offer certain advantages, it does not constitute a fundamentally new contribution.

2. The setting of the parameter $\lambda$ is crucial to the effectiveness of the proposed method. In similar works, such as Zhang et al., 2023a and Xu et al., 2023, the authors provide thorough discussions on the selection and impact of $\lambda$. In contrast, this paper lacks sufficient theoretical or empirical discussion on the choice of $\lambda$, offering only marginal experimental insights. A more detailed analysis of how $\lambda$ influences the algorithm's performance, along with a broader range of experiments, would significantly strengthen the work.

3. In the References section, the entries for Zhang et al., 2023a and Zhang et al., 2023b appear to be identical, which seems to be an error. The authors should carefully review and correct this duplication to ensure the accuracy and integrity of the references.

**Questions For Authors:**

Please refer to the above comments.

**Relation To Broader Scientific Literature:**

The key contributions are closely related to the broder scientific literature.

**Theoretical Claims:**

The authors claim that the convergence rate of OPSA is independent of the condition number of the target matrix $X_*$. However, in Theorem 5.4, they assume that $\sigma_r(X_*)=1$, which implies that ${\kappa} (X_*)={||X||}$ (the operator norm of matrix $X_*$). This assumption suggests that the convergence results are inherently tied to $||X_*||$, creating an apparent contradiction with the claim of condition number independence. Specifically, if the convergence rate depends on $||X_*||$, it indirectly depends on the condition number $\kappa(X_*) $, given the assumption $\sigma_r(X_*)=1$. This discrepancy warrants further clarification from the authors to reconcile the theoretical claims with the assumptions made in the analysis.

---

> ### Author Rebuttal · Authors · 2025-03-31
>
> We would like to thank the reviewer for their effort to review our paper and for the constructive feedback! We address all reviewers' comments/concerns below.
>
> > Summary: The paper is well-written, with clear and coherent exposition, and presents compelling results that contribute meaningfully to the field.
>
> We would like to thank the reviewer for recognizing the significance of our contribution and the positive evaluation of our work!
>
> > Claims And Evidence
> >> The authors assert that  ....
>
> Following the reviewers’ suggestion we have accordingly updated Figure 4.  The updated Figures can be found [here](https://ibb.co/WpH1yZvq) and [here](https://ibb.co/YBcJSmd4), and show linear convergence to ground truth up to the order of $10^{-13}$ (not exactly zero relative error due to numerical limits and roundoff errors).
>
>
> >> Furthermore, the results in Figure 4 indicate ....
>
> We have carried out additional experiments considering higher values of condition numbers (i.e., $\kappa=1000, 10000$). The new empirical results can be found [here](https://ibb.co/rKJhrFn3) and [here](https://ibb.co/svZ9Qnv0), and show that our proposed algorithm still converges linearly even for significantly larger condition numbers and more challenging scenarios.
>
> >Theoretical Claims
>
> We believe there is a misunderstanding here. As noted in Remark 5.16, the initialization condition does depend on the condition number of $\kappa(X_\ast)= || X_\ast ||_{op}$.
>
> However, the **rate of convergence** provided in Theorem 5.4 is
>  $1 - \frac{0.12}{\chi^2}$  and it is **independent of $\kappa(X_\ast)$**. We have further emphasized this distinction in the revised version of our paper.
>
> > Experimental Designs Or Analyses
>
> >> Regarding the experimental results in Figure 5...
>
> Please see our response above and the updated figures provided in the anonymous links therein.
>
> >> Additionally, the authors state ...
>
> Thank you for this comment! Indeed, the statement we made was incorrect and we have accordingly revised it. An extremely large $\lambda$ will significantly slow down convergence but not lead to local minima if the initialization condition given in Theorem 5.4 is satisfied.
> We verified this claim in the additional experiments we conducted and have included them in the revised paper. In the updated figures, which can be found [here](https://ibb.co/G32nRDnY) and [here](https://ibb.co/KJhrpYS), we observe that OPSA  converges linearly for a higher value of $\lambda$ but now much slower, requiring $\approx 10000$ iterations to converge.
>
> > Other Strengths And Weaknesses
>
> >> The authors claim that the distance metric introduced in Eq. (22) is novel.
>
> We would like to point out that the distance metric itself is a key ingredient that led to the technical contributions needed for proving Theorem 5.4. Specifically, this new distance metric renders the previous proof techniques used in Tong et al., 2021a, not applicable in this overparameterized rank setting that we address in this paper. We would like to refer the reviewer to our detailed responses above on a similar concern raised by Reviewers Uzmr and oy8C.
>
> >> The setting of the parameter $\lambda$ ...
>
> As reported in our numerical results (see Figure 5), OPSA is emperically robust to a wide selection of $\lambda$, showcasing good performance and similar iteration complexity for a range of values from $10^{-4}$ to $2$. Theoretically speaking, from our existing theoretical results, we can easily derive a lower bound for $\lambda$, i.e., $\lambda \geq c$ with $c$ being some universal positive constant. On the other hand, an extremely large $\lambda$ will empirically and theoretically slow down the convergence, as we have discussed above, also see the updated figures ( [here](https://ibb.co/G32nRDnY) and [here](https://ibb.co/KJhrpYS)) and Theorem 5.4. By setting a desired convergence rate, an upper bound of $\lambda$ can also be computed, which resembles the bound in Xu et al, 2023. That said, to choose a mild $\lambda$ that works with OPSA is not difficult in many cases. We have added a short discussion on this in the revised version of the paper.
>
> >> In the References ...
>
> Thank you for pointing this out! We have fixed this issue in the revised version of our paper.

---

### Official Review · Reviewer_oXqM · 2025-03-13

**Overall Recommendation:** 3

**Summary:**

In "Guarantees of a Preconditioned Subgradient Algorithm for Overaparametrized Asymmetric Low-rank Matrix Recovery" the authors provide a novel method to solve robust asymmetric and overparametrized matrix sensing problems without the rate scaling with the condition number of the solution matrix. The authors provide numerical and theoretical evidence of the algorithm's performance.

**Claims And Evidence:**

Main claim: The proposed algorithm converges linearly when applied to robust and overparametrized matrix sensing problems. Further, the convergence rate is independent of the ground truth matrix condition number.

The authors establish that the claim is true in theorem 5.4 and provide numerical evidence that even with overparametrization, the convergence rate does not degrade as abruptly as that of existing schemes.

**Essential References Not Discussed:**

N/A

**Experimental Designs Or Analyses:**

Yes, I do not think the experimental design has issues.

**Methods And Evaluation Criteria:**

Yes.

**Other Comments Or Suggestions:**

N/A

**Other Strengths And Weaknesses:**

Weaknesses:
1) No bounds for RIP constants (and consequently restricted smoothness and sharpness) are provided for the Gaussian case.
2) Requires spectral initialization (but this is standard in the robust case)
3) Rate dependence on the restricted condition number of the loss, which may be dimension dependent.
4) sensitivity to hyperparameter lambda which requires a lot of knowledge to be tuned appropriately.

**Questions For Authors:**

Q1) The authors introduced the mixed norm RIP. Can the authors compare their notion of RIP to those existing in the literature? Other notions of RIP have been introduced in the robust setting.
Q2) Is it possible to establish guarantees for the RIP for Gaussian matrices? As is, it is tested empirically. It would be good to know whether the proposed notion of RIP scales with increased d. The provided simulations only provide results for fixed rank 10, while the restricted smoothness and sharpness must hold for matrices of rank d.

**Relation To Broader Scientific Literature:**

The authors suitably position their work in the literature.

**Theoretical Claims:**

I did not check the correctness of the proofs.

---

> ### Author Rebuttal · Authors · 2025-03-31
>
> We would like to thank the reviewer for taking the time to review our paper and the valuable feedback! Next, we provide point-by-point responses to all reviewer’s comments and concerns.
>
> > Other Strengths and Weaknesses
>
> >> No bounds for RIP constants (and consequently restricted smoothness and sharpness) are provided for the Gaussian case.
>
> Assuming a measurement operator with Gaussian i.i.d. $\frac{1}{p}\mathcal{N}(0,1)$ for the matrix sensing problem, we can invoke the result of Tong et al 2021, which gives the following bounds for the RIP constants:
> - $\delta^{-}_{2d} \gtrsim 1$
> - $\delta^{+}_{2d} \lesssim 1$
> - $\delta_0 \gtrsim 1 - 2p_s$
>
> as long as  the sample complexity satisfies  $p \gtrsim \frac{(m+n)d}{(1-2p_s)^2}\mathrm{log}( \frac{1}{1-2p_s})$, where $p_s$ is the outlier's ratio. A short discussion on this has been added to the revised paper.
>
> >> Requires spectral initialization (but this is standard in the robust case)
>
> Indeed! Our work, similarly to other SOTA robust matrix sensing works, e.g. Tong et al 2021, requires spectral initialization. For future work, random initialization is one of the tasks we will try to tackle for robust matrix sensing.
>
> >>  Rate dependence on the restricted condition number of the loss, which may be dimension dependent.
>
> You are right. The rate of convergence depends on $\chi$ which is the condition number of the loss. Unfortunately, to the best of our knowledge, this dependence of $\chi$ is something that all current SOTA works in this area share. We will work on removing this dependence in future works.
>
> >>  Sensitivity to hyperparameter lambda which requires a lot of knowledge to be tuned appropriately.
>
> As shown in the experimental section, the performance of our algorithms is quite robust to a wide selection of $\lambda$. For details, we refer the reviewer to Figure 5. Note, however, that a massive $\lambda$ value may slow down the convergence, as it matches our theory (Please see also our [response](https://openreview.net/forum?id=GaCo82yC7z&noteId=hMHh5h5Ohj) to Reviewer Uzmr on this).
>
>
> > Questions For Authors:
>
> >> Q1) The authors introduced the mixed norm RIP. Can the authors compare their notion of RIP to those existing in the literature? Other notions of RIP have been introduced in the robust setting.
>
> Thank you for this question. The mixed-RIP norm has been actually used in prior works e.g. Tong et al 2021. We refer the reviewer to our [response](https://openreview.net/forum?id=GaCo82yC7z&noteId=2UnKEoGVze) to Reviewer oy8C, and [Char21], which shows the analytical derivation of this mixed-RIP condition.
>
> >> Q2) Is it possible to establish guarantees for the RIP for Gaussian matrices? As is, it is tested empirically. It would be good to know whether the proposed notion of RIP scales with increased d. The provided simulations only provide results for fixed rank 10, while the restricted smoothness and sharpness must hold for matrices of rank d.
>
> We would like to refer the reviewer to our previous response above on this topic. Indeed, the sample complexity bound now scales linearly with $d$ instead of $r$. Please also note that in the synthetic experiments, the true rank $r$ was always overestimated with $d\geq r$. Our reported results show linear convergence in the overaparameterized rank regime, validating our theory.

---

### Official Review · Reviewer_Uzmr · 2025-03-13

**Overall Recommendation:** 4

**Summary:**

The paper presents a preconditioned subgradient method for robust low-rank matrix sensing using a Burer-Monteiro factorization, focusing on the case where the rank $r$ of the ground truth signal is not known. The preconditioner used is a straightforward modification of the preconditioner proposed by [Xu et al.](https://arxiv.org/abs/2302.01186) tailored to the asymmetric case. The authors show that the resulting subgradient method, when initialized near the solution set, converges to a global minimizer at a geometric rate independent of the condition number of the unknown signal and present favorable numerical results on synthetic problems.

**Claims And Evidence:**

The main claim in the paper -- namely, geometric convergence independent of the condition number -- is supported both theoretically and experimentally. I have a few issues and questions related to other claims made in various places in the submission:

- In Remark 5.6, the authors state that "the initialization condition is negatively affected as this condition number increases" (ostensibly referring to the condition number of $X_{\star}$). However the initialization condition only depends on the spectral norm of $X_{\star}$, which can be kept constant as the condition number increases. Is there a typo in the statement Theorem 5.4? Please clarify.

- The sentence in Line 434 (left column) reads: "if the $\lambda$ parameter is set too large, OPSA may get stuck at some local minimum". I suspect this interpretation of the numerical evidence is wrong -- in particular, Theorem 5.4 imposes an initial condition of $\mathrm{dist}(F_0, F^{\star}) \leq \lambda \epsilon$, which is *milder* as $\lambda$ increases. My understanding, based on Theorem 5.4, is that larger $\lambda$ simply push the contraction factor $\rho(\chi, \delta, \epsilon)$ closer to 1 leading to very slow convergence rather than stagnation. Please clarify this point as well.

**Essential References Not Discussed:**

Most essential references are already discussed. The idea of RIP under "$\mathcal{I}$-outlier bounds", which appears without attribution, can be attributed to [this paper](https://link.springer.com/article/10.1007/s10208-020-09490-9) as well as [this earlier work](https://arxiv.org/abs/1705.02356).

**Experimental Designs Or Analyses:**

No issues with experimental designs or analyses.

**Methods And Evaluation Criteria:**

The method is evaluated on synthetic problem instances, which is standard in the literature on matrix sensing. Experiments on non-synthetic benchmarks, especially regarding the tuning of the preconditioner's $\lambda$ parameter, could strengthen the paper but are not necessary.

**Other Comments Or Suggestions:**

- Please take care to remove unused notation (e.g., $\otimes$ and $\mathrm{vec}$).
- Please introduce necessary notation as needed. For example $\Delta_{L}$ and $\Delta_{R}$ in the appendix are used before they are defined.
- Possible typos in (2) and (3): it should be $L_t^{\mathsf{T}} L_t$ instead of $L_t L_t^{\mathsf{T}}$. Similarly for $R_t$. Moreover, you should be taking the gradient of some loss function which is missing.
- Possible typo in Corollary 5.10: should the iteration complexity depend on $\delta_{2d}^+$ rather than $\delta_{2r}^+$?
- Possible typo in Proposition 5.12 / Corollary 5.13: should $L$ be $\delta_{2d}^+$ instead of $\delta_{2r}^+$?
- A discussion of the sample complexity of your method, especially in the presence of outliers, would be useful.
- More broadly: can the dependence be improved from $\delta_{2d}^+$ to $\delta_{d+r}^{+}$?

**Other Strengths And Weaknesses:**

Strengths:

- The problem and algorithm are clearly motivated.
- The numerical experiments cover a variety of regimes (albeit all being synthetic).

Weaknesses:

- Given the discussion under "Relation To Broader Scientific Literature", the main contribution of this paper is to cover the $\ell_1$ loss with outliers given an asymmetric Burer-Monteiro factorization. With appropriate book-keeping (e.g.,, defining the correct distance measure -- see also my next comment below), the "restricted regularity" framework followed by the authors is well-tread by analyses (Tong et al., 2021).
- I disagree with the claim (under "Technical Innovation") that the distance metric in (22) is novel. It is already known from the work of (Tong et al. 2021) and (Cheng & Zhao, 2024) that the distance must be measured in the norm induced by the preconditioner (and that, for that purpose, it suffices to use the norm induced by the preconditioner evaluated at the optimal solution - this leads precisely to the norm used in (22)). It is possible that I am missing something here, and I welcome any clarification.
- (Relatively minor weakness): The paper appears to have been written (or revised) somewhat hastily. There are several typos, unused notation (e.g., the Kronecker product symbol) and passages that can be polished. For example, nothing about the "low-rank matrix estimation problem" defined in Eq. (5) suggests that the solution should be low-rank!

**Questions For Authors:**

N/A

**Relation To Broader Scientific Literature:**

The authors have done an adequate job positioning the paper within the broader literature on matrix sensing. However:

- The proposed algorithm is essentially the same as the one in [Cheng & Zhao](https://ieeexplore.ieee.org/document/10446187). In particular, both algorithms treat the loss function as the composition of a well-conditioned "outer" loss with a poorly-conditioned "inner mapping", and apply the exact same preconditioner to the subdifferential/gradient of the "outer" loss. This work is cited in the paper but omitted from Table 1, while it covers the asymmetric case with unknown rank and the resulting convergence rate is independent of $\kappa(X_{\star})$.

- In Table 1, the ScaledGD($\lambda$) method appears misattributed to (Xiong et al. 2023), when (to my knowledge) it was proposed in the work of (Xu et al., 2023).

**Theoretical Claims:**

I am familiar with the literature on (overparameterized) matrix sensing and the proof outline makes sense to me. However, I only checked the proofs at a high level. I believe the theoretical claims contain some typos, which I will list under the "Other comments or suggestions" portion of the review.

---

> ### Author Rebuttal · Authors · 2025-03-31
>
> We appreciate the reviewer's time in thoroughly evaluating our paper and for providing insightful comments and feedback. Below, we provide point-by-point responses addressing each of the reviewer’s comments and concerns.
>
> > Claims And Evidence
>
> We are glad that the reviewer has found our main claim well-supported both theoretically and experimentally!
>
>  >> In Remark 5.6,  ...
>
> Thank you for this question. Please note that in the statement of Theorem 5.4 we say that $\sigma_r(X_*)=1$, hence the condition number $\kappa(X_\ast)$ is equal to the spectral norm of $X_\ast$  i.e., $\kappa(X_\ast) = \frac{\sigma_1(X_\ast)}{\sigma_r(X_\ast)} = \sigma_1(X_\ast) = ||X_\ast||_{op}$.
>
> >> The sentence in Line 434 ...
>
> Thank you for bringing up this point! Indeed, the wording in the paper “***OPSA may get stuck at some local minimum"***  is not accurate and we have revised it. You are right to say that a large $\lambda$ will relax the initialization condition, but slow down convergence. We verified this claim in the additional experiments we conducted, which are included in the revised paper. As observed in the updated figures, which can be found [here](https://ibb.co/G32nRDnY) and [here](https://ibb.co/KJhrpYS), OPSA still converges linearly for a higher value of $\lambda=10$ but now much slower than in the case of lower values of $\lambda$, requiring $\approx$ 10000 iterations to converge.
>
> >Methods And Evaluation Criteria
>
> To strengthen the experimental section, we provide additional experiments on robust matrix completion on real datasets, which can be found [here](https://ibb.co/5XVfRNQf) and [here](https://ibb.co/hJsTH8qm). The results showcase the merits of our approach in this real-world application and have been included in the revised version of our paper. Please see also our [responses](https://openreview.net/forum?id=GaCo82yC7z&noteId=2UnKEoGVze) to Reviewer oy8C.
>
> > Theoretical Claims
>
> Thank you for your time to review our proofs and for finding them reasonable. In the revised version of the paper, we have fixed the listed typos and some others!
>
> > Relation To Broader Scientific Literature
>
> Following your suggestion, we have added Cheng&Zhao’s [C&Z] to Table 1. Indeed C&Z's approach addresses the asymmetric and overparameterized setting, however, they focus on smooth losses, unlike our work which addresses non-smooth loss functions. We should also note that C&Z reports linear convergence in function values unlike our work, which relies on a distance metric based on updates of matrix factors.
>
> >> In Table 1, the ScaledGD() method appears misattributed ...
>
>  We have fixed that in the revised manuscript.
>
> > Essential References Not Discussed
>
> That’s correct! We have added the references for these two papers to the revised manuscript.
>
> > Other Strengths And Weaknesses
>
> >>Strengths
>
> Thank you for your positive evaluation of our work!
>
> >> Weaknesses
>
> >> Given the discussion under ...
>
> Note that  Tong et al. 2021 does not address the overparameterized rank setting. Our contribution generalizes their approach to include this setting, leading to non-trivial technical insights. Please refer to our [response](https://openreview.net/forum?id=GaCo82yC7z&noteId=2UnKEoGVze) to reviewer oy8C and the next comment for further details.
>
> >> I disagree with the claim ...
>
> Thank you for bringing up this point! It is true that  ***the norm induced by the preconditioner evaluated at the optimal solution*** is exactly what we need to use as the distance metric. However, please note that Tong2021 uses non-regularized preconditioners, hence, our distance is a generalization of that one. Moreover, to the best of our knowledge, C&Z shows linear convergence in function values and it is not clear in the paper that the same distance metric as ours is being used at all. Note also that the distance is not the main contribution of the paper, but rather the key point for the technical innovation. In particular, demonstrating linear convergence and contraction for this distance metric is far from a trivial extension of prior work (e.g., Tong, 2021). The bounds and matrix inequalities established in earlier studies cannot be directly applied to this more general, perturbed version of the distance metric.
>
> >> (Relatively minor weakness)
>
> We have fixed all these issues in the revised manuscript!
>
> > Other Comments Or Suggestions
> >> Sample Complexity Discussion
>
> Assuming a measurement operator with Gaussian i.i.d. $\mathcal{N}(0,1)$ for the matrix sensing problem we can invoke the result of Tong et al 2021, which gives the following sample complexity bound for the mixed-RIP and $\mathcal{S}$-outlier bound conditions to hold w.h.p. $p \gtrsim \frac{(m+n)d}{(1-2p_s)^2}\mathrm{log}( \frac{1}{1-2p_s})$, where $p_s$ is the outlier's ratio. A short discussion on this has been added to the revised paper.

---

> > ### Comment · Reviewer_Uzmr · 2025-04-04
> >
> > Thank you for your answers. I have revised my score upwards.
> >
> > > However, please note that Tong2021 uses non-regularized preconditioners, hence, our distance is a generalization of that one [...]
> >
> > I do not dispute that the technical points in your analysis might be nontrivial to carry out. The point that I was making is that the induced metric in which you measure convergence is "standard" once a preconditioner has been chosen, and regularized preconditioners are certainly not new in the overparameterized matrix sensing / factorization literature.
> >
> > > To strengthen the experimental section, we provide additional experiments on robust matrix completion on real datasets
> >
> > Thanks for adding these. Does your theory imply any guarantees for matrix completion? As far as I know, the operator $P_{\Omega}$ in matrix completion does not satisfy the restricted isometry property without additional assumptions.

---

> > > ### Author Response · Authors · 2025-04-06
> > >
> > > Thank you very much for raising the score, and we appreciate that you recognize our contributions.
> > >
> > > >Does your theory imply any guarantees for matrix completion? As far as I know, the operator in matrix completion does not satisfy the restricted isometry property without additional assumptions
> > >
> > > You are right that the sampling operator $P_{\Omega} $ doesn't hold RIP condition in the sense $||\frac{1}{p}P_\Omega-\mathbf{I}||$ is not uniformly bounded by a small constant (at least not with high probability) under the operator norm. Note that $||(\frac{1}{p}P_\Omega-\mathbf{I})X||\leq \epsilon$ holds for a fixed matrix $X$, but the independence requirement destroys the RIP and makes it almost useless for iterative convergence analysis. Thus, our theorem doesn't directly guarantee the convergence of matrix completion, just like most, if not all, matrix sensing convergence theorems.
> > >
> > > The RIP for matrix completion is usually presented in the form of $||P_T-\frac{1}{p}P_T P_\Omega P_T||\leq\epsilon$, where $P_T$ is a projection onto the tangent space $T$ of the low-rank manifold at the groundtruth. It holds with high probability if the groundtruth is $\mu$-incoherent, and of course, subject to the sampling pattern of $\Omega$. That said, our proofs in this paper can be used as a blueprint for the matrix completion case. We anticipate that the proof can be worked out by following our blueprint, but the technical details will be more involved as the RIP is presented in a different way.
> > >
> > > We hope this answers your question. We also encourage other reviewers to have follow-up discussions with us. If you are happy with our rebuttals/answers, please adjust your scores accordingly. Thank you very much.

---

### Official Review · Reviewer_oy8C · 2025-03-18

**Overall Recommendation:** 3

**Summary:**

This paper studies the problem of recovering a low-rank matrix from noisy linear measurements of the matrix (i.e. inner products with the vectorization of the matrix). This paper studies the problem at a particular level of generality:
- We have adversarial noisy measurements of an ill-conditioned asymmetric matrix and want to recover it by solving a minimizing over all low-rank matrices with respect to a non-smooth loss function.

If we remove any of these requirements (noisy, ill-conditioned, asymmetric, non-smooth loss), then this problem has been solved by prior work.

They show that a generalization of prior algorithms succeed in recovering low-rank matrices in this setting with an algorithm that converges exponentially quickly (i.e. has an iteration complexity that's logarithmic in the desired precision).

Paper is mostly theory with some lightweight experiments.

**Claims And Evidence:**

The core claims of the paper are convincing at a glance. I have no particular issues with the claims in the paper, but I also didn't invest the time to dig into their theorems.

The paper seems to argue a simple generalization of prior techniques to get a slightly more general result that previously possible using a very believable algorithm that just sorta seems like the right generalization of prior works.

The empirics are nice, but not sufficient to make me feel comfortable recommending this algorithm to practitioners. Paper is mostly theory, but that feels mostly solid. Some mild gaps in what they discuss, but nothing severe.

In short, I'm comfy accepting the paper.

**Essential References Not Discussed:**

None in particular.

**Experimental Designs Or Analyses:**

N/A (basically, this is the same as "Methods and Evaluation Criteria" imo)

**Methods And Evaluation Criteria:**

The paper has an empiric section, but it's somewhat lacking.

They propose this simple method for low-rank matrix recovery, but only apply it to the simplest toy case where our underlying ground truth matrix is the product of "two $n \times r$ random matrices" [Line 381, left], we make iid Gaussian measurements of our matrix, and the noise is randomly positioned with uniform at random (admittedly very large) values.

This is a very nice starting point for analyzing an algorithm, the sorta sanity check you want that this method actually works as theorized, but doesn't show that this method works at all for anything more realistic. I would really have wanted to see a real-world matrix that's operated upon. Perhaps a matrix completion or a robust PCA application, as they point out as potential applications in the first paragraph of their introduction [Lines 42-48, left]?

Further, the authors only use one algorithm as a benchmark, despite the fact that they cite many prior works as algorithms that try to solve low-rank matrix recover (see e.g. table 1 on page 3).

The empirics are good enough to make me believe that the proposed algorithms has the potential to work, but is not enough to make me believe that the algorithm actually works well.

I'll note that I'm by no means an expert in the space of low-rank matrix recovery, so I'm not sure what the best benchmarks are, but something involving real world matrices should be possible. **I would love to hear if other reviewers have other benchmarks in mind.** Plus, given that they already have some experiments running, conceptually shouldn't be so hard to get running on other real-world matrices.

**Other Comments Or Suggestions:**

This is a list of typos and recommended edits. Ignore whatever you want to ignore.

1. [Eqn 5] This general optimization problem is fine, but should it optimize over rank(X) \leq something?
2. [Line 172, left] Idk if the language of A as a measurement matrix is standard. If so, ignore more. But if not, then maybe call it a linear map from R^{m by n} to R? Or call it a matrix, but apply A to vec(X*)? Or specify that A_i(X*) = <A_i, X*> via the trace inner product?
3. [Line 193, right] Remark 4.1 is great, glad it's included!
4. [Line 258, left] Naively substituting terms in, we're assuming that $\lambda = \frac1{20}$.
5. [Eqn 15] I think this is supposed to be an assumption on the initial iterate F0, not an implication from the definition of $\varepsilon$? Needs to be rewritten here to clarify this.
6. [Thm 5.4] What is bar lambda? I see that lambda is the regularization inside the preconditioner, but idk what bar lambda is.
7. [Thm 5.4] Add a line which says something like "In particular, $t = O(\chi^2 \log(c_\chi \sqrt{\|\|X_*\|\|}))$ iterations sufficed to have ||L_t R_t' - X_*||_F \leq \eta dist(F0, F*)", or whatever the a good final guarantee is.
8. [Thm 5.4] Why assume WLOG that the r^th singular value is 1? Instead, just write the condition number of X instead of the operator norm. It feels more honest imo, unless there's another reason to normalize this way in the body of the paper.
9. [Thm 5.4] What is C_\chi?
10. [Line 253, right] Is "liner rate" a typo?
11. [Line 255, right] What rate in 5.4 is this? Is this the expresson on the RHS of dist(Ft, F*)? Of ||Lt Rt' - X*||?
12. [Line 280, left] Wonderful! Chi grows with d! Can we make this more rigorous with a specific example?
13. [Defintion 5.8] So here, ||X||_F is the L2 norm of vec(X). And A(x) is a vector whos entries are inner products with vec(X). So, ||A(X)||_1 = ||B*vec(X)||_1 for some matrix B that depends on A. Then, we see this mixed-norm RIP property is the exact same as the 2->1 norm of the matrix B [See e.g. "Estimating the matrix p → q norm" by Guth et al]. Not sure this is interesting or superbly relevant, but it's an interesting connection to a term in classical numerical linear algebra.
14. [Fig 1] I think this! Very simple and nice visual! Write out what the blue and red values actually are maybe? Just a ballpark is fine! Or their ratio, for the sake of Prop 5.9!
15. [Fig 2] Also very nice! Simple clean story about OPSA being slower with overparameterization, but much less vulnerable than ScaledSM!
16. [Corols 5.10, 5.13] You've overloaded the meaning of $\varepsilon$ between here and Thm 5.4. I suggest you change $\varepsilon$ in Thm 5.4 to like $\Delta$ or $\eta$ or just something else. Also, you need to acknoledge the (logarithmic) dependence on the quality of the starting iterate via like C_\chi and \kappa.
17. [Section 5.4] Add some words before sending me straight into a definition. For instance, tell me a bit bout the model of noise you use, and if it's inspired by any prior work's formulation. Also, tell me what Definition 5.11 is really trying to encode.
18. [Figures 2 and 3] These figures seem to have basically the same information, just slightly different input parameters. Replace one of these.

**Other Strengths And Weaknesses:**

The writing in the paper is really nice. Like I'm super enjoying the fact that, as I read the paper, I'll wonder "oh, but does this require us to know the value of the loss function at the true optima" and the paper will have a remark saying something like "in practice, we don't know this value of the loss function at the true optima, so we use XYZ in practice instead and that works great", and I greatly enjoy this. This pre-empting of the natural questions, and having good answers to these questions, made this paper particularly fun to read.

I will say they dropped the ball a little bit in section 5.4, where they almost entirely lack this flavor text.

**Questions For Authors:**

What's the recommended way to handle not knowing the true rank r a priori? Doubling? There a goodness of fit metric to concern outselves with?

What interesting loss functions fit assumptions 5.1 and 5.2, but not the non-rank-restricted versions of these loss functions? I.e. how helpful is it to assume that X1, X2, X are rank <= d?

What's going on with Eqn 15? Is this supposed to be a theorem proved about F0, or an assumption on the quality of our first iterate?

Remark 5.6 points out that the condition number does matter for the initial iterate's assumption. This should be acknowledged sooner.

What does Defn 5.11 really try to encode? This is a weird expression to read in Equation 17. We compare the L1 to L2 norm as done in Defn 5.8, but there's this splitting of A across S and the complement of S, but the norm is outside of the minus sign. It's all a bit odd at a glance.

Lines 370-373 point out that Corol 5.13 shows the rate of convergence depends on properties of the measurement matrix. Why is this mentioned here? Wasn't this also the case in Corol 5.10?

**Relation To Broader Scientific Literature:**

This is maybe my sticking point for this paper.

The results seem essentially correct. The experiments seem fine. The improvement over the prior work is that this paper proposes a way to satisfy 4 directions of generality as the same time in low-rank matrix recovery:
1. (Bounded) Adversarial noise
2. Asymmetric matrix
3. Ill-conditioned matrix
4. Non-smooth loss function

The proposed algorithm is intuitively very natural following from how the authors describe the prior work. I don't want to undersell the fact that it does require novelty, but isn't like a wild idea or anything either. It's more like figuring out the right way to generalize the prior ideas.

This makes me think about that annoying word "marginal". This work is proposing an algorithm that's theoretically capable in a very general setting, but hasn't been empirically demonstrated to work amazingly on real-world data (it's all very synthetic data). So I can't sell the importance of the paper on it's empirical prowess, it's instead got to mostly rely on its theoretical prowess instead. The generality is nice, but just how big of a deal is it? Not sure, and I think that at the end of the day it does cross the line to being significant enough. But it is a bit of a toss-up to me.

**Theoretical Claims:**

I didn't check the details of the proofs at any stage. I just followed the vague intuitions throughout the paper.

Nothing here seems like a really bold surprising claim. Their main theoretical result (theorem 5.4) is a pretty reasonable-looking claim, saying that if we start from a good initialization point then their iterative method converges to an extremely good solution. They also say, perhaps a bit too vaguely, that a prior work gives an easy method to produce a good-enough starting point.

I have some minor peeves with the way the paper writes some parts of it's theory (such as sorta obfuscating the fact that theorem 5.4 has a mild dependence on conditioning, which is totally cool and fine, but obfuscated). I exhaust my little peeves in my list of typos and recommended edits. I don't really have concerns about the theoretical correctness broadly though.

---

> ### Author Rebuttal · Authors · 2025-03-31
>
> We thank the reviewer for taking the time to carefully review our paper and for providing such valuable comments and constructive feedback. Below, we present our detailed, point-by-point responses to each of the reviewer’s comments and concerns.
>
> > Claims & Evidence
>
> Thank you for your positive words and for finding our theoretical results to be solid. Our  contributions are mainly theoretical, and hence, the experimental part  focused on corroborating the theoretical results. However, the proposed algorithm can also offer significant benefits in addressing real-world problems! Please see further details below on additional experiments conducted on real data.
>
> > Methods & Evaluation Criteria
>
> To address reviewers’ concerns, we provide additional results on video background subtraction, which is a classic application for robust matrix completion.  We test with two well-known video datasets and compare the performance of the proposed OPSA to a state-of-the-art robust PCA method, which requires exact rank and full observation. The visual results can be found [here](https://ibb.co/5XVfRNQf) and [here](https://ibb.co/hJsTH8qm), showing that OPSA is competitive against the exact-rank robust PCA algorithm, despite that the former is given a rough overestimate of the rank and 30% of the observations.
>
> > Theoretical Claims
>
> Thank you for your positive evaluation of our theoretical claims! In fact, Theorem 5.4 shows a linear rate of convergence that is ***independent of the condition number***. However, the **initialisation condition does indeed depend on the condition number (as also noted in Remark 5.6)**. We will further emphasize this point in the revised version of our paper!
>
> > Relation To Broader Scientific Literature
>
> Again, we would like to thank the reviewer for recognising the novelty of our theoretical contributions.  Our work essentially addresses an open problem in the low-rank matrix factorization literature. As the reviewer recognizes,  theoretical challenges led to non-trivial technical contributions. Our experimental section was designed to corroborate the theoretical results, e.g., showing a linear rate of convergence in the rank overparameterized regime, etc. We understand that this might have caused a misunderstanding as to the practicality of the proposed algorithms. We hope that the additional real-world experiments we provided (please see details above) addressed the reviewer's concerns.
>
> > Other Comments/Suggestions
>
> In the revised paper, we have fixed typos and provided clarification based on comments/suggestions! Below, we analytically respond to some of the key points raised here:
>
> 6 -$\bar{\lambda}$ is an auxiliary variable used for the sake of deriving a simple form for the rate of convergence.  $\bar{\lambda}$ shows up in the expression we use for $\lambda$ i.e., $\lambda =\frac{\|X_\ast\|_{op}}{c \bar{\lambda}}$. This definition of $\lambda$ ameliorates the derivation of the rate of convergence given in Theorem 5.4.
>
> 8 - The reason for normalizing this way is to provide some useful insights into the rate of convergence. As also mentioned above, the rate of convergence does not depend on the condition number, however, the initialization condition does depend on it.
>
> 9 - This is $c\cdot \chi$, we will add $\cdot$ to make it clear.
>
> 11 - This is the general expression for the rate for any $\lambda$ for the RHS of $\mathrm{dist}(\mathbf{F},\mathbf{F}_\ast)$.
>
> > Questions For Authors: - Handling of not knowing the true rank a priori.
>
> In practical settings a safe overestimate of true rank but not too large $d$  would work! Of course, this initial guess is problem-dependent and requires prior domain knowledge. Note also that OPSA is not too sensitive to the selection of $d$ (please see Fig. 2), allowing for a range $d < 2r$ without paying a significant price, when it comes to the speed of convergence.
>
> > What interesting loss functions fit Assumps 5.1 & 5.2...
>
> Convex losses combined with linear maps that satisfy RIP (e.g., our objective for the matrix sensing problem) would satisfy Assumptions 5.1 and 5.2 for restricted rank but not for non-rank-restricted versions. The reason for that is that RIP properties would normally hold for low-rank $X$.
>
> > What's going on with (15)?...
>
> This is just an assumption! We have revised the statement to make this clear.
>
> > What does Def 5.11 really try to encode?...
>
> The S-outlier bound property has been used in the robust low-rank matrix recovery problem in prior works, e.g., (Tong et al, 2021a). It actually encodes a property of $\mathcal{A}$ that allows restricted sharpness condition to be satisfied in matrix sensing problems in the presence of outliers. We would like to refer the reviewer to a detailed derivation of this condition, as a natural generalization of RIP can be found in [Char21].
>
>  [Char21] Charisopoulos, et al. "Low-rank matrix recovery with composite optimization: good conditioning and rapid convergence." Found. of Comput. Math., 2021.

---

### Decision · Program_Chairs · 2025-05-01

**Decision:**

Accept (poster)

**Comment:**

This paper proposes an Overparameterized Preconditioned Subgradient Algorithm (OPSA) for recovering low-rank asymmetric matrices under gross corruption, with theoretical guarantees of linear convergence independent of the matrix’s rank and condition number. The paper makes meaningful theoretical contributions and extends existing techniques to a more general, previously unresolved setting: overparameterized, asymmetric, ill-conditioned, and robust matrix sensing with non-smooth loss.

The primary theoretical result (Theorem 5.4) establishes linear convergence for OPSA with a convergence rate that is independent of the condition number. Reviewers Uzmr and oXqM recognized this as a substantial and nontrivial advance, especially in the overparameterized asymmetric setting. On the other hand, Reviewers uniformly noted that the original empirical section was too limited, relying solely on synthetic data and offering limited comparative analysis. The authors partially addressed this by adding experiments on real datasets (robust matrix completion) and analyzing convergence behavior under larger condition numbers and lower relative error tolerances. However, Reviewer oy8C still sees room for a more systematic comparison (e.g., error vs. time plots) to further validate OPSA’s practical performance.

Based on the reviews and the authors' thorough rebuttal, I recommend acceptance, as the contribution is theoretically solid and the paper has been improved substantially through the rebuttal process.